# Robust and Consistent Ski Rental with Distributional Advice

**Jihwan Kim** [1]   **Chenglin Fan** [1]

## Abstract

The ski rental problem is a canonical model for online decision-making under uncertainty, capturing the fundamental trade-off between repeated rental costs and a one-time purchase. While classical algorithms focus on worst-case competitive ratios and recent "learning-augmented" methods leverage point-estimate predictions, neither approach fully exploits the richness of full distributional predictions while maintaining rigorous robustness guarantees. We address this gap by establishing a systematic framework that integrates distributional advice of unknown quality into both deterministic and randomized algorithms. For the *deterministic setting*, we formalize the problem under perfect distributional prediction and derive an efficient algorithm to compute the optimal threshold-buy day. We provide a rigorous performance analysis, identifying sufficient conditions on the predicted distribution under which the expected competitive ratio (ECR) matches the classic optimal randomized bound. To handle imperfect predictions, we propose the *Clamp Policy*, which restricts the buying threshold to a safe range controlled by a tunable parameter. We show that this policy is both *robust*, maintaining good performance even with large prediction errors, and *consistent*, approaching the optimal performance as predictions become accurate. For the *randomized setting*, we characterize the stopping distribution via a *Water-Filling Algorithm*, which optimizes expected cost while strictly satisfying robustness constraints. Experimental results across diverse distributions (Gaussian, geometric, and bi-modal) demonstrate that our framework improves consistency by significantly over existing point-prediction baselines while maintaining comparable robustness.

## 1. Introduction

The ski rental problem stands as a paradigmatic example of online decision-making under uncertainty, encapsulating the fundamental trade-off between short-term incremental costs (renting) and long-term fixed investments (buying). Since its formalization, it has served as a cornerstone for analyzing online algorithms, with applications spanning resource allocation, subscription services, equipment leasing, and cloud computing, where decisions must be made sequentially without full knowledge of future demand or usage duration. In the classic setting, the optimal deterministic strategy achieves a competitive ratio of 2 (Karlin et al., 1988), while randomized algorithms improve this bound to $\frac{e}{e-1} \approx 1.58$ (Karlin et al., 1994). However, these traditional approaches rely on worst-case analysis, ignoring potentially valuable information about the distribution of the usage duration (i.e., the number of skiing days). With the advent of machine learning and predictive modeling, leveraging distributional predictions of future outcomes has emerged as a powerful avenue to narrow the gap between online and offline performance.

While recent work has explored integrating point predictions into ski rental and related online problems (Purohit et al., 2018), point predictions fail to capture the uncertainty inherent in real-world scenarios. A single predicted value cannot account for the variability of possible usage durations, leading to suboptimal decisions when the true duration deviates significantly from the prediction. In contrast, distributional predictions—by quantifying the probability of all possible usage durations—enable more nuanced and robust decision-making. For instance, knowing that the probability of skiing for 1 day is 0.8 and for 5 days is 0.2 (as in our concrete example) allows us to compute a threshold that balances the risk of over-renting and over-buying, which a point prediction (e.g., the expected value of 1.8 days) cannot achieve. This observation motivates our focus on distributional prediction for the ski rental problem: we aim to design policies that explicitly leverage the full distribution of usage duration to minimize expected cost, while providing theoretical guarantees on performance even when predictions are imperfect.

---

[1]Seoul National University, Seoul 08826, South Korea. Emails: aqua4689@snu.ac.kr, fanchenglin@gmail.com. Correspondence to: Chenglin Fan <fanchenglin@gmail.com>.

*Proceedings of the $43^{rd}$ International Conference on Machine Learning*, Seoul, South Korea. PMLR 306, 2026. Copyright 2026 by the author(s).

## 1.1. Contributions and Overview

This paper makes several core contributions to the study of the ski rental problem under distributional prediction:

First, we formalize the distributional ski rental setting with perfect prediction, focusing on threshold-based policies (rent for $t - 1$ days, buy on day $t$). We derive the expected cost function for such policies, propose an efficient $O(n)$ algorithm to compute the optimal threshold $t^*$ (given a bounded support of the duration distribution), and analyze the expected competitive ratio (ECR) relative to the offline optimum (which knows the true duration in advance). We establish exact ECR expressions and distribution-dependent upper bounds for two key cases ($t \leq b$ and $t > b$, where $b$ is the buy cost) and identify sufficient conditions for the ECR to match or over the optimal randomized bound of $\frac{e}{e-1}$. A concrete example demonstrates that our distributional approach outperforms point prediction, achieving a lower expected cost by adapting to the structure of the duration distribution.

Second, we address the critical challenge of prediction uncertainty. In practice, predicted distributions ($\hat{p}$) often deviate from the true distribution ($p$), and policies that over-rely on imperfect predictions risk catastrophic performance. To mitigate this, we introduce a **clamp policy** that restricts the optimal threshold from the predicted distribution ($t_{\hat{p}}^*$) to an interval $[\lceil \lambda b \rceil, \lfloor b/\lambda \rfloor]$ (for $\lambda \in (0,1)$). We prove that this policy achieves two desirable properties: (1) **robustness**: the ECR is bounded by $1 + \frac{1}{\lambda} - \frac{1}{b}$ regardless of prediction error (measured via 1-Wasserstein distance), ensuring performance does not degrade arbitrarily with prediction quality; and (2) **consistency**: as the prediction error vanishes, the ECR approaches the optimal value for the true distribution, leveraging accurate distributional information when available. We further extend these results to the total variation distance.

Third, we generalize our analysis to randomized algorithms, which are known to outperform deterministic strategies in worst-case ski rental. We formulate necessary robustness constraints for randomized policies, deriving conditions on the cumulative distribution function (CDF) of the duration that ensure a bounded ECR. For monotonic cost functions, we show that the optimal CDF follows an exponential form. We then propose an iterative algorithm with binary search, inspired by a "water filling" interpretation of resource allocation, ensuring the robustness constraints are satisfied while minimizing expected cost.

Experimental Validation: We evaluate our approach across a range of representative predicted distributions—including uniform, discretized Gaussian, truncated geometric, and bi-modal—comparing against corrected baselines adapted from (Purohit et al., 2018) for distributional inputs. Our method consistently improves consistency by roughly 5–20% over the baselines, with the largest gains observed on bi-modal and geometric distributions, where uncertainty is most impactful. Importantly, these improvements are achieved while maintaining the same robustness guarantees. These results demonstrate that our framework effectively balances robustness and consistency across diverse distribution types.

## 1.2. Related Work

The ski rental problem is a canonical model in online competitive analysis, capturing the fundamental rent-or-buy dilemma. Early work established the optimal deterministic 2-competitive break-even strategy (Karlin et al., 1988) and the optimal randomized $\frac{e}{e-1}$-competitive algorithm (Karlin et al., 1994), laying the foundation for subsequent work on online algorithms, and extended to related settings such as dynamic TCP acknowledgment (Karlin et al., 2003). These guarantees were later derived and unified via primal–dual techniques (Buchbinder & Naor, 2009).

Learning-augmented algorithms have achieved notable success in online optimization, starting with the seminal work of Lykouris and Vassilvitskii (Lykouris & Vassilvtiskii, 2018) and follow-ups (Purohit et al., 2018; Rohatgi, 2020; Azar et al., 2021; Bansal et al., 2022; Lindermayr & Megow, 2022; Angelopoulos et al., 2020). Recent advances extend these techniques to offline tasks such as matching, clustering, and sorting (Dinitz et al., 2021; Ergun et al., 2021; Bai & Coester, 2023; Lu et al., 2021), as well as to data structures (Lin et al., 2022), robust predictors (Anand et al., 2022; Antoniadis et al., 2023), binary search with distributional predictions (Dinitz et al., 2024), warm-start strategies (Blum & Srinivas, 2025), dynamic graphs (Brand et al., 2024), and more. Learning-augmented algorithms have been studied for the classical rent-or-buy problem using point prediction. Purohit et al. (2018) gave both deterministic and randomized algorithms, later shown to be optimal by Angelopoulos et al. (2020), Bamas et al. (2020), and Wei & Zhang (2020). The classical ski rental problem has been extended to settings with multiple predictions and multiple shops (Gollapudi & Panigrahi, 2019; Angelopoulos et al., 2020; Bamas et al., 2020; Wei & Zhang, 2020; Banerjee, 2020; Wang et al., 2020; Shin et al., 2023; Li et al., 2025), capturing richer online decision-making scenarios. studied a ski-rental variant with the number of days drawn from a distribution with known moments.

**Distributional Predictions.** The integration of distributional information into online decision-making has evolved significantly over the past decade. Early work by Khanafer et al. (2013) incorporated limited statistical information by assuming knowledge of only low-order moments (such as the mean or variance) of the arrival distribution. Using a

game-theoretic, zero-sum framework, they derived robust randomized algorithms; however, their approach is restricted to coarse statistical inputs and cannot leverage the detailed, full-distribution predictions generated by modern machine learning models. More recently, a growing body of literature has begun investigating true "distributional predictions," where an advisor provides a full probability distribution rather than a single point estimate. This paradigm was initiated by Dinitz et al. (2024) for the binary search problem and has since been expanded to other fundamental online domains, such as metric matching (Canonne et al., 2025) and online bidding (Angelopoulos & Simon, 2025).Within the specific context of the ski rental problem, other models leveraging distributions exist but operate under fundamentally different threat models and objectives. For instance, the sampling-based stochastic model of Diakonikolas et al. (2021) assumes drawing i.i.d. samples from an unknown but honest underlying distribution to minimize expected cost. Meanwhile, Kang et al. (2025) adopt a Bayesian framework with full discrete priors, updating posterior beliefs over time to maximize expected utility. A closely related setting was recently explored by Besbes et al. (2025), who studied the ski rental problem with full distributional predictions. However, their analysis assumes that the algorithm knows an upper bound on the distance between the predicted and true distributions in advance, leaving the setting with an unknown error bound as an open question. Concurrent to our work, Cui & Dinitz (2026) addressed this question by providing an algorithm that achieves bounded expected additive loss without knowing the prediction error. While these methods allow for rich uncertainty modeling, they assume the ground truth is drawn from a true underlying distribution or prior, and they do not explicitly optimize or guarantee worst-case competitive ratios against an adversarial stopping time.

Our work bridges this gap by providing a systematic treatment of the canonical ski rental problem under an untrusted-advisor framework with full distributional advice. Unlike settings that assume a true underlying distribution, we evaluate our algorithm's performance against the actual realized stopping time, ensuring that the provided distribution can originate from a potentially inaccurate or maliciously misleading source. This allows us to explicitly optimize the expected competitive ratio against realized worst-case instances, thereby providing rigorous robustness guarantees."

### 1.3. Organization

The paper systematically develops a framework for the ski rental problem, beginning with theoretical foundations and progressing to practical validation. Section 2 formalizes the problem, introduces key notation, and establishes baseline cost models. Section 3 analyzes the perfect prediction scenario, deriving optimal threshold policies and exact competi-

itive ratio expressions, illustrating how distributional predictions improve over point predictions. Section 4 introduces the clamp policy to handle prediction errors, proving robustness and consistency under various distance metrics. Section 5 generalizes the framework to randomized algorithms, deriving optimal distributions and proposing an iterative optimization procedure. Finally, Section 6 presents experimental validation across diverse distributions, showing that our approach consistently outperforms adapted baselines, particularly on bi-modal and geometric cases.

## 2. Preliminaries and Problem Setup

### 2.1. The Ski Rental Problem

The classical ski rental problem models an online decision-making task with an unknown time horizon. An agent repeatedly decides whether to rent skis at a cost of 1 per day or buy skis once at a fixed cost $b > 1$. The total number of skiing days $D \in \mathbb{N}$ is unknown to the agent in advance. If the agent buys skis on day $t$, then it incurs a total cost of $(t - 1) + b$; if it never buys, its cost is $D$.

An online algorithm is evaluated via its competitive ratio (CR), defined as the worst-case ratio between the algorithm's cost and that of an optimal offline strategy that knows $D$ in advance. The offline optimum incurs cost $\mathrm{OPT}(D) = \min\{D, b\}$.

### 2.2. Distributional Predictions

We consider a setting in which the algorithm is given a distributional prediction over the unknown horizon $D$. Specifically, before any decisions are made, the algorithm observes a distribution $\mathcal{D}$ over $D$. The distribution may be inaccurate or adversarially chosen, but it provides probabilistic information about the future.

An algorithm may leverage $\mathcal{D}$ to randomize its buying time or to adaptively decide whether to buy. Importantly, we distinguish between consistency and robustness. Consistency refers to performance guarantees when $D \sim \mathcal{D}$ is indeed drawn from the predicted distribution, while robustness refers to worst-case guarantees that hold for any realization of $D$, regardless of $\mathcal{D}$.

### 2.3. Algorithms and Randomization

A (possibly randomized) online algorithm $\mathcal{A}$ selects a buying time $\tau \in \{1, 2, \dots\} \cup \{\infty\}$, where $\tau = \infty$ corresponds to never buying. The cost incurred by $\mathcal{A}$ on horizon $D$ is

$$\mathsf{cost}_{\mathcal{A}}(D) = \begin{cases} D, & \text{if } \tau > D, \\ (\tau - 1) + b, & \text{if } \tau \leq D. \end{cases}$$

For randomized algorithms, the cost is defined in expecta-

tion over the internal randomness of $\mathcal{A}$.

## 2.4. Performance Metrics

We use two related but distinct notions of performance. First, because our advice is distributional, we evaluate consistency under a distribution over the skiing horizon. Given a distribution $\mathcal{D}$ over $\mathbb{N}$, we define the expected competitive ratio under $\mathcal{D}$ as

$$\text{ECR}_{\mathcal{A}}(\mathcal{D}) := \frac{\mathbb{E}_{D \sim \mathcal{D}}\left[\mathbb{E}_{\mathcal{A}}\left[\text{cost}_{\mathcal{A}}(D)\right]\right]}{\mathbb{E}_{D \sim \mathcal{D}}\left[\text{OPT}(D)\right]}.$$

Here the inner expectation is taken over the internal randomness of $\mathcal{A}$.

For robustness, we use the classical worst-case competitive ratio. For a realized skiing horizon $d \in \mathbb{N}$, define the pointwise competitive ratio

$$r_{\mathcal{A}}(d) := \frac{\mathbb{E}_{\mathcal{A}}\left[\text{cost}_{\mathcal{A}}(d)\right]}{\text{OPT}(d)}.$$

The worst-case competitive ratio is then

$$\text{WCR}_{\mathcal{A}} := \sup_{d \in \mathbb{N}} r_{\mathcal{A}}(d) = \sup_{d \in \mathbb{N}} \frac{\mathbb{E}_{\mathcal{A}}\left[\text{cost}_{\mathcal{A}}(d)\right]}{\text{OPT}(d)}.$$

## 2.5. Notation

Throughout the paper, we use $\mathbb{P}[\cdot]$ to denote probability, $\mathbb{E}[\cdot]$ for expectation, and OPT for the offline optimal cost. All logarithms are natural unless stated otherwise.

## 3. Perfect Prediction Setting

### 3.1. Algorithm

Let $D \in \mathbb{N}$ be a random variable denoting the total number of skiing days, and let its distribution be

$$p(d) = \Pr[D = d].$$

The per-day rental cost is normalized to 1, and the purchase cost is $b$. We consider *threshold policies* that rent skis for the first $t - 1$ days and buy on day $t$. Let $\mathcal{A}_t$ denote the threshold algorithm that buys on day $t$.

The corresponding expected cost is

$$g(t) = \mathbb{E}_{D \sim p}\left[\text{cost}_{\mathcal{A}_t}(D)\right]$$
$$= \sum_{d=1}^{t-1} p(d)\, d \;+\; \sum_{d=t}^{\infty} p(d)\,(b + t - 1).$$

Our objective is to determine

$$t^* = \arg \min_{t \in \mathbb{N} \cup \{\infty\}} g(t),$$

and to purchase ski on day $t^*$.

Assuming a known upper bound `max_day` on the support of $D$, a simple $O(\texttt{max\_day})$-time routine for computing $t^*$ is given below.

---

**Algorithm 1** Optimal Threshold for Ski Rental with Perfect Prediction

---

**Require:** Probability distribution $p[1..\texttt{max\_day}]$, buy cost $b$, upper bound `max_day`
**Ensure:** Optimal threshold $t^*$ minimizing expected cost $g(t)$
1: Initialize $\text{min\_cost} \leftarrow \infty, t^* \leftarrow 1$
2: Precompute tail probabilities $S(t) = \sum_{d=t}^{\infty} p[d]$:
3: $\text{tail}[\texttt{max\_day} + 1] \leftarrow 0$
4: **for** $d = \texttt{max\_day}, \ldots, 1$ **do**
5:     $\text{tail}[d] \leftarrow \text{tail}[d + 1] + p[d]$
6: **end for**
7: Initialize $\text{rent\_cost} \leftarrow 0$ {This tracks $\sum_{d=1}^{t-1} p[d] \cdot d$}
8: **for** $t = 1, \ldots, \texttt{max\_day} + 1$ **do**
9:     $\text{buy\_cost} \leftarrow \text{tail}[t] \cdot (b + t - 1)$
10:    $g(t) \leftarrow \text{rent\_cost} + \text{buy\_cost}$
11:    **if** $f_t < \text{min\_cost}$ **then**
12:        $\text{min\_cost} \leftarrow g(t)$
13:        $t^* \leftarrow t$
14:    **end if**
15:    **if** $t \leq \texttt{max\_day}$ **then**
16:        $\text{rent\_cost} \leftarrow \text{rent\_cost} + p[t] \cdot t$
17:    **end if**
18: **end for**
19: **return** $t^*$

---

### 3.2. A Concrete Example with $1 < t^* < \infty$

This example illustrates a case where our setting provides an advantage compared to a point prediction with perfect information. For the point prediction, $t^*$ is 1 or $\infty$. We introduce a case where $1 < t^* < \infty$ holds, thereby having an advantage compared to point prediction.

Assume

$$p(d) = \begin{cases} 0.8, & d = 1, \\ 0.2, & d = 5, \\ 0, & \text{otherwise}, \end{cases} \qquad b = 3.$$

Then, for $t = 1, 2, \ldots$, we compute

$$g(t) = \sum_{d=1}^{t-1} p(d)\, d + \sum_{d=t}^{\infty} p(d)\,(b + t - 1).$$

In particular:

Hence, the unique minimizer is

$$\boxed{t^* = 2, \quad 1 < t^* < \infty.}$$

| $t$ | $\sum_{d<t} p(d)\, d$ | $\sum_{d\geq t} p(d)$ | $b+t-1$ | $g(t)$ |
|---|---|---|---|---|
| 1 | 0 | 1.0 | 3 | 3.0 |
| 2 | $0.8 \cdot 1 = 0.8$ | 0.2 | 4 | 1.6 |
| 3 | $0.8 \cdot 1 = 0.8$ | 0.2 | 5 | 1.8 |
| 4 | $0.8 \cdot 1 = 0.8$ | 0.2 | 6 | 2.0 |
| 5 | $0.8 \cdot 1 = 0.8$ | 0.2 | 7 | 2.2 |
| $\geq 6$ | $0.8 \cdot 1 + 0.2 \cdot 5 = 1.8$ | 0 | — | 1.8 |

*Table 1.* Example computation of $g(t)$ for discrete distribution.

### 3.3. Performance Analysis

We define the survival function $S(t) := \sum_{d \geq t} p(d)$ and the expected offline optimum:

$$\mathrm{OPT}(p) = \mathbb{E}_{D \sim p}[\mathrm{OPT}(D)] = \sum_{d<b} p(d)d + b \cdot S(b).$$

Under the perfect prediction setting, where the algorithm knows $p$ and selects the optimal threshold $t^*$, we denote the resulting threshold policy as $\mathcal{A}_{t^*}$. We abbreviate its expected competitive ratio $\mathrm{ECR}_{\mathcal{A}_{t^*}}(p)$ as:

$$\mathrm{ECR}(p) = \frac{f(t^*)}{\mathrm{OPT}(p)}.$$

Our analysis distinguishes two regimes depending on whether $t^*$ lies before or after the buy cost $b$.

**Theorem 3.1** (Competitive Ratio for $t^* \leq b$). *If the optimal threshold satisfies $t^* \leq b$, the $\mathrm{ECR}(p)$ is bounded by:*

$$\mathrm{ECR}(p) \leq 1 + \frac{(b-1)r - (b-t^*)}{t^* r + (b-t^*)}, \quad \text{where } r = \frac{S(t^*)}{S(b)} \geq 1.$$

**Theorem 3.2** (Competitive Ratio for $t^* > b$). *If the optimal threshold satisfies $t^* > b$, the $\mathrm{ECR}(p)$ is bounded by:*

$$\mathrm{ECR}(p) \leq \frac{t^* - 1}{b} + r, \quad \text{where } r = \frac{S(t^*)}{S(b)} \in [0, 1].$$

Two bounds show that the competitive ratio is governed by how much probability mass lies in the interval $[t^*, b)$, captured by the ratio $r$.

### 3.4. Sufficient Conditions for $C$-Consistency

Let $C > 1$ and $\alpha = t^*/b$. Define the tail ratio

$$r(\alpha) := \frac{S(t^*)}{S(b)} = \frac{S(\alpha b)}{S(b)}.$$

Note that $r(\alpha) \geq 1$ when $\alpha \leq 1$ and $r(\alpha) \in [0, 1]$ when $\alpha \geq 1$.

**Theorem 3.3.** *A threshold policy with threshold $t^* = \alpha b$ is $C$-Consistent if either of the following holds:*

*1. **Early Buy:** $\alpha \leq 1$ and*

$$r(\alpha) \leq \frac{C(1-\alpha)}{1 - \frac{1}{b} - (C-1)\alpha}.$$

*2. **Late Buy:** $\alpha \geq 1$ and*

$$r(\alpha) \leq C - \alpha + \frac{1}{b}.$$

Achieving $C$-Consistency requires either that little probability mass lies between $t^*$ and $b$ in the early-buy regime (so that $S(t^*)$ is not much larger than $S(b)$), or that the tail beyond $b$ decays sufficiently fast in the late-buy regime (so that $S(t^*)$ is small relative to $S(b)$).

In particular, to compare against the ratio with no prediction, one may simply substitute $C = \frac{e}{e-1}$.

## 4. Robustness under Distributional Shift

In practical scenarios, the predicted distribution $\hat{p}$ may deviate from the true distribution $p$. We quantify this prediction error using the 1-Wasserstein distance

$$\eta := W_1(p, \hat{p})$$

with ground metric $|i - j|$. To achieve robustness against large errors while remaining consistent with accurate predictions, we introduce the *Clamp Policy*.

### 4.1. The Clamp Policy

Let $t_{\hat{p}}^* \in \arg\min_t g_{\hat{p}}(t)$ denote the optimal threshold for the predicted distribution. For a fixed hyperparameter $\lambda \in (0, 1)$, the clamped threshold $\tilde{t}$ is defined as follows:

**Definition 4.1** (Clamp Policy). Given prediction $\hat{p}$ and robustness parameter $\lambda$, the algorithm buys ski on day

$$\tilde{t} := \min\Big\{\max\{t_{\hat{p}}^*, \lceil \lambda b \rceil\}, \lfloor b/\lambda \rfloor\Big\}.$$

This policy guarantees that even in the presence of extremely poor predictions, the threshold remains within a safe range relative to the buying cost $b$.

### 4.2. Case Study: $\lambda = 1/3$

Consider a scenario with $\lambda = 1/3$:

- Suppose skiing $2b/3$ days occurs with probability $1/2$ and skiing $2b$ days occurs with probability $1/2$.

- **Optimal Threshold:** $t^* = 2b/3 + 1$, i.e., buy skis after $2b/3$ days.

**Expected Cost Calculation:**

$$\mathrm{Cost} = \frac{1}{2}\left(\frac{2}{3}b\right) + \frac{1}{2}\left(\frac{2}{3}b + b\right) = \frac{7}{6}b.$$

For comparison, deterministic policy in Purohit et al. (2018) (buying at $b/3$ or $3b$) would yield $\mathrm{Cost} = \frac{4}{3}b$.

### 4.3. Stability and Distribution-Free Bounds

The following two lemmas provide the stability and robustness bounds used in our analysis.

**Lemma 4.2** (Wasserstein Stability). *Let $\eta := W_1(p, \hat{p})$, where the ground metric is $|d - d'|$. Then, for every $t \in \mathbb{N}$,*

$$\mathbb{E}_{D \sim p}[\mathsf{cost}_{\mathcal{A}_t}(D)] \leq \mathbb{E}_{D \sim \hat{p}}[\mathsf{cost}_{\mathcal{A}_t}(D)] + b\eta,$$

*and*

$$\mathbb{E}_{D \sim p}[\mathrm{OPT}(D)] \geq \mathbb{E}_{D \sim \hat{p}}[\mathrm{OPT}(D)] - \eta.$$

**Lemma 4.3** (Distribution-Free Threshold Bound). *For every $t \in \mathbb{N}$,*

$$\mathrm{WCR}_{\mathcal{A}_t} \leq \begin{cases} 1 + \dfrac{b-1}{t}, & t \leq b, \\[2mm] 1 + \dfrac{t-1}{b}, & t \geq b. \end{cases}$$

*Consequently, the same bound holds for $\mathrm{ECR}_{\mathcal{A}_t}(q)$ under every distribution $q$.*

### 4.4. Robust-Consistent Guarantee

We now state the main guarantee of the Clamp Policy.

**Theorem 4.4** (Robust-Consistent Bound). *Let $\tilde{t}$ be the threshold selected from the prediction $\hat{p}$, and let*

$$\mathcal{A}_{\mathrm{Clamp}} := \mathcal{A}_{\tilde{t}}.$$

*Then*

$$\mathrm{WCR}_{\mathcal{A}_{\mathrm{Clamp}}} \leq 1 + \frac{1}{\lambda} - \frac{1}{b}.$$

*Moreover, let*

$$\eta := W_1(p, \hat{p}),$$

*and assume that*

$$\eta < \mathbb{E}_{D \sim \hat{p}}[\mathrm{OPT}(D)].$$

*Define*

$$\theta := \frac{\eta}{\mathbb{E}_{D \sim \hat{p}}[\mathrm{OPT}(D)]} \in [0, 1).$$

*Then*

$$\mathrm{ECR}_{\mathcal{A}_{\mathrm{Clamp}}}(p) \leq \frac{\mathrm{ECR}_{\mathcal{A}_{\mathrm{Clamp}}}(\hat{p}) + b\theta}{1 - \theta}.$$

*Consequently,*

$$\mathrm{ECR}_{\mathcal{A}_{\mathrm{Clamp}}}(p) \leq \min \left\{ 1 + \frac{1}{\lambda} - \frac{1}{b}, \frac{\mathrm{ECR}_{\mathcal{A}_{\mathrm{Clamp}}}(\hat{p}) + b\theta}{1 - \theta} \right\}. \tag{1}$$

**Corollary 4.5** (Robustness and Consistency). *The Clamp Policy satisfies*

$$\mathrm{WCR}_{\mathcal{A}_{\mathrm{Clamp}}} \leq 1 + \frac{1}{\lambda} - \frac{1}{b}$$

*regardless of the prediction quality.*

*Furthermore, fix a predicted distribution $\hat{p}$ and the corresponding clamped threshold $\tilde{t}$. As*

$$W_1(p, \hat{p}) \to 0,$$

*we have*

$$\mathrm{ECR}_{\mathcal{A}_{\mathrm{Clamp}}}(p) \longrightarrow \mathrm{ECR}_{\mathcal{A}_{\mathrm{Clamp}}}(\hat{p}).$$

*In particular, when $p = \hat{p}$, the performance under the true distribution is exactly the predicted performance of the clamped policy.*

*Remark* 4.6 (Alternative Metric). Suppose instead that prediction error is measured by total variation:

$$\eta_{\mathrm{TV}} := \frac{1}{2} \|p - \hat{p}\|_1.$$

Then

$$\mathbb{E}_{D \sim p} \left[ \mathsf{cost}_{\mathcal{A}_{\mathrm{Clamp}}}(D) \right] \leq \mathbb{E}_{D \sim \hat{p}} \left[ \mathsf{cost}_{\mathcal{A}_{\mathrm{Clamp}}}(D) \right] + \eta_{\mathrm{TV}}(b + \tilde{t} - 1),$$
$$\mathbb{E}_{D \sim p}[\mathrm{OPT}(D)] \geq \mathbb{E}_{D \sim \hat{p}}[\mathrm{OPT}(D)] - b\eta_{\mathrm{TV}}.$$

Therefore, whenever

$$\mathbb{E}_{D \sim \hat{p}}[\mathrm{OPT}(D)] > b\eta_{\mathrm{TV}},$$

we have

$$\mathrm{ECR}_{\mathcal{A}_{\mathrm{Clamp}}}(p) \leq \frac{\mathrm{ECR}_{\mathcal{A}_{\mathrm{Clamp}}}(\hat{p}) + \dfrac{\eta_{\mathrm{TV}}(b + \tilde{t} - 1)}{\mathbb{E}_{D \sim \hat{p}}[\mathrm{OPT}(D)]}}{1 - \dfrac{b\eta_{\mathrm{TV}}}{\mathbb{E}_{D \sim \hat{p}}[\mathrm{OPT}(D)]}}.$$

The worst-case bound

$$\mathrm{WCR}_{\mathcal{A}_{\mathrm{Clamp}}} \leq 1 + \frac{1}{\lambda} - \frac{1}{b}$$

is unchanged because it depends only on the range of $\tilde{t}$, not on the metric used to measure prediction error. The proof is deferred to Appendix B.4.

## 5. Randomized Algorithms

Deterministic threshold policies provide a robust safety net, but their competitive performance is fundamentally limited: without any prediction, no deterministic algorithm can achieve a competitive ratio better than 2 (Karlin et al., 1988). Randomization allows for improved guarantees; the classical randomized bound achieves a competitive ratio of $e/(e-1) \approx 1.58$ (Karlin et al., 1994).

In this section, we generalize from a single deterministic threshold to *randomized stopping distributions*. Our objective is to minimize the expected cost under a predicted distribution $\hat{p}$ while enforcing a hard robustness guarantee $R$.

## 5.1. Equivalent Conditions for Robustness

Let $x \in \mathbb{N}$ be the length of the skiing day, and let $z \in \mathbb{N}$ be the buying day, and define $C_z(x)$ as the cost. Then $C_z(x) = x$ if $z > x$, and $C_z(x) = b + z - 1$ if $z \leq x$.

A randomized stopping policy $f$ is $R$-robust if $\mathbb{E}_{z \sim f}[C_z(x)] \leq R \cdot \mathrm{OPT}(x)$ for all $x \in \mathbb{N}$. Define

$$F(x) = \sum_{t=1}^{x} f(t), \qquad \mu(x) = \sum_{t=1}^{x} (t-1)\, f(t).$$

**Lemma 5.1** (Robustness Constraints). *A randomized stopping policy $f$ is $R$-robust if and only if the following two conditions are satisfied.*

*1. For all $1 \leq x \leq b - 1$,*

$$\mu(x) + (b-x)F(x) \leq (R-1)x.$$

*2. The following holds.*

$$\mu(\infty) = \sum_{t=1}^{\infty} (t-1)f(t) \leq (R-1)b.$$

## 5.2. Optimization of the Stopping Distribution

Given a predicted distribution $\hat{p}$ over skiing days, the consistency cost for buying on day $t$ is

$$g(t) = \sum_{d<t} \hat{p}(d)d + \sum_{d \geq t} \hat{p}(d)(b + t - 1). \qquad (2)$$

The optimization problem is

$$\min_f \sum_{z=0}^{\infty} g(z)f(z)$$

s.t. the robustness constraints in Lemma 5.1.

**Theorem 5.2.** *If $g$ is monotone increasing,*

$$F^*(x) = \min\left((R-1)\left(\left(1 + \frac{1}{b-1}\right)^x - 1\right), 1\right)$$

*is an optimal distribution.*

**Theorem 5.3** (Extension beyond monotone $g$). *Assume that $g$ satisfies the following condition: there exists an integer $y \geq b - 1 + \ln(R/(R-1))/\ln(b/(b-1))$ such that*

*1. $g(1) \leq g(2) \leq \cdots \leq g(y)$,*

*2. $g(t) \geq g(y + 1 - b)$ for all $t \geq y + 1$.*

*Then the geometric CDF $F^*$ given in Theorem 5.2 is still optimal. In particular, this covers the one-hot prediction case $p(y) = 1$ with $y \geq b - 1 + \ln(R/(R-1))/\ln(b/(b-1))$, where $g(t) = b + t - 1$ for $t \leq y$ and $g(t) = y$ for $t \geq y + 1$.*

*Remark* 5.4. This geometric structure is a natural extension of the randomized ski rental algorithm of Purohit et al. (2018). For the one-hot prediction $y$, if $y \geq b - 1 + \ln(R/(R-1))/\ln(b/(b-1))$, we can apply Theorem 5.3, and it leads to the same structure as the randomized ski rental algorithm of Purohit et al. (2018). However, if $y < b - 1 + \ln(R/(R-1))/\ln(b/(b-1))$, the structure of the randomized ski rental algorithm of Purohit et al. (2018) is suboptimal, and we provide an optimal distribution in Theorem C.1.

## 5.3. The Water-Filling Algorithm

Reformulating (2), the consistency cost $g(t)$ of stopping at day $t$ is

$$g(t) = \left(\sum_{d \geq t} \hat{p}(d)\right)t + \left(\sum_{d < t} \hat{p}(d)d + \sum_{d \geq t} \hat{p}(d)(b-1)\right). \tag{3}$$

Therefore, for a predicted distribution $\hat{p}$ with finite support $\{d_1, \ldots, d_n\}$, the consistency cost $g(t)$ of stopping at day $t$ is a piecewise linear function over the intervals $(d_k, d_{k+1}]$.

**Algorithm Intuition.** We assume a certain level $h$. We fill the probability mass on the area $\{x : g(x) \leq h\}$. We allocate the maximum probability as we can while satisfying the robustness constraint. If we cannot allocate the total probability mass 1, then the level $h$ is infeasible, and we have to increase $h$. We want to find a minimum feasible $h$ to minimize the cost $\sum_{z=0}^{\infty} g(z)f(z)$ while allocating total mass 1. We find it approximately by binary search.

An implementation-level description is provided in Appendix D.7.

### High-Level Algorithm Strategy.

1. Calculate the cost function $g$

2. Initialize the interval of height and set $h$ as a midpoint

3. For the interval $[0, b-1]$, fill weights maximally using the update rules

4. For the interval $[b, \infty)$, check the feasibility

5. If feasible, reduce to the lower interval; otherwise, reduce to the upper interval.

## 6. Experimental Evaluation

In this section, we present experimental results for our proposed algorithm.[1]

---

**Algorithm 2** Water-Filling for Randomized Robust Ski Rental

**Require:** Prediction $p$, buy cost $b$, robustness $R$
**Ensure:** Probability Mass Function $f$
1: Compute piecewise linear cost function $g(t) = a_k t + b_k$.
2: Perform binary search for water level $h$:
3:   For each segment $(d_i, d_{i+1}]$ with $g(d_i) \leq h$ and $d_i < b$, find endpoint

$$e_i = \min \left\{ \left\lfloor \frac{h - b_i}{a_i} \right\rfloor, d_{i+1}, b \right\}.$$

4: Going through active segments, calculate $F(x)$ as following.
5:   Update $F(x)$ along $(d_i, e_i]$ using geometric growth from the tightness condition:

$$F(x + 1) = F(x) + \frac{F(x) + R - 1}{b - 1}.$$

6:   For the starting point $d_j$ of next support ($j > i$ is a minimum number with $g(d_j) \leq h$), assign mass

$$m = \frac{(R - 1 + F(e_i))(d_j - e_i)}{b - 1},$$

$$F(d_j) = F(e_i) + m.$$

7:   Check feasibility at the tail: If there is $d_k \geq b$ satisfying $g(d_k) \leq h$ and

$$d_k \leq \frac{(R - 1)b - \mu(b - 1)}{1 - F(b - 1)},$$

then $h$ is feasible. Otherwise, $h$ is infeasible, and we have to consider a bigger $h$.
8:   Adjust $h$ via binary search to find the minimum feasible $h$ approximately.

## 6.1. Consistency Comparison under Fixed Robustness

**Classical Ski Rental.**  We find a relation between the robustness parameter $R$ and the parameter $\lambda$ used in Purohit et al. (2018). For the randomized algorithm of Purohit et al. (2018), the competitive ratio is given by $R = \frac{1 + \frac{1}{b}}{1 - e^{-(\lambda - \frac{1}{b})}}$. Rearranging the expression yields

$$\lambda = \frac{1}{b} - \log \left( 1 - \frac{1 + \frac{1}{b}}{R} \right). \quad (4)$$

This expression allows us to directly map our robustness parameter $R$ to the parameter $\lambda$ of Purohit et al. (2018) for experimental comparison.

**Consistency Metric.**  For a ski day length distribution $p$, define the cost function

$$g(t) = \sum_{d < t} p(d)d + \sum_{d \geq t} p(d)(b + t - 1), \quad t \geq 1.$$

For a *randomized* buying day $Z \in \mathbb{N}$, the consistency is defined as

$$\mathrm{Cons}(p) := \frac{\mathbb{E}[g(Z)]}{\min_{t \in \mathbb{N}} g(t)}. \quad (5)$$

Although we use the notation $\mathrm{Cons}(p)$ for the experimental evaluation, this is not a new performance notion separate from the expected competitive ratio defined in Section 2.4. It is a normalized version of ECR. Indeed, let $\mathcal{A}_Z$ denote the randomized threshold policy that draws a buying day $Z$ and then buys on day $Z$. Then

$$\mathrm{ECR}_{\mathcal{A}_Z}(p) = \frac{\mathbb{E}[g(Z)]}{\mathrm{OPT}(p)}.$$

Similarly, for the best deterministic threshold policy under $p$,

$$\min_{t \in \mathbb{N}} \mathrm{ECR}_{\mathcal{A}_t}(p) = \frac{\min_{t \in \mathbb{N}} g(t)}{\mathrm{OPT}(p)}.$$

Therefore,

$$\mathrm{Cons}(p) = \frac{\mathbb{E}[g(Z)]}{\min_{t \in \mathbb{N}} g(t)} = \frac{\mathrm{ECR}_{\mathcal{A}_Z}(p)}{\min_{t \in \mathbb{N}} \mathrm{ECR}_{\mathcal{A}_t}(p)}.$$

Thus $\mathrm{Cons}(p)$ measures the multiplicative loss in ECR relative to the best threshold policy for the same distribution $p$. This normalization preserves the same ordering as ECR.

**Purohit et al. (2018)-style Baselines under Distributional Predictions.**  To make a comparison fair, we fix $(b, R)$ and let $\lambda$ as (4).

Algorithm 3 in Purohit et al. (2018) assumes a *point prediction* $y$ and chooses one of two branch distributions depending on whether $y \geq b$ or $y < b$. When the predictor outputs a distribution $p$ with $y \sim p$, we adapt the branch condition as follows.

Let $P := \Pr[y \geq b]$ under the predicted distribution. We consider two baselines for producing a randomized buying day $Z$:

- **Majority-branch baseline:** use the $y \geq b$ branch if $P > 1/2$, otherwise the $y < b$ branch.

- **Mixture baseline:** mix the two branch distributions proportionally to $P$:

$$\Pr[Z = j] = P \cdot q_j + (1 - P) \cdot r_j, \quad (6)$$

  where $q$ and $r$ are the distributions of Algorithm 3 for the $y \geq b$ and $y < b$ branches, respectively.

In all cases (ours and baselines), (5) is evaluated using the same $g(\cdot)$ derived from the skiing day distribution $p$.

*Table 2.* Consistency comparison (smaller is better) at $(b, R) = (50, 1.7)$.

| $p$ family | **Ours** | Purohit (maj) | Purohit (mix) |
|---|---|---|---|
| unif100 | **1.1612** | 1.1782 | 1.1866 |
| unif200 | **1.3331** | 1.3492 | 1.3643 |
| gauss | **1.3375** | 1.4195 | 1.4169 |
| geom | **1.2879** | 1.4114 | 1.4183 |
| twopoint | **1.0415** | 1.2448 | 1.2547 |

**Experimental Setup.** We set $(b, R) = (50, 1.7)$ and consider five representative skiing day distributions $p$ that allocate nontrivial probability mass on both sides of $b$:

1. Uniform on $\{1, \ldots, 100\}$ (unif100)

2. Uniform on $\{1, \ldots, 200\}$ (unif200)

3. Discretized Gaussian $\mathcal{N}(50, 12^2)$ truncated to $\{1, \ldots, 150\}$ (gauss)

4. Truncated geometric with parameter $0.05$ on $\{1, \ldots, 600\}$ (geom)

5. Two-point distribution $0.7\delta_{30} + 0.3\delta_{120}$ (twopoint)

For our method, we compute the policy via water-filling and bisection, then construct the PMF of $Z$ and evaluate $\mathbb{E}[g(Z)]$. For both baselines, we compute their PMF of $Z$ based on branching and evaluate the $\mathbb{E}[g(Z)]$.

**Discussion.** Table 2 demonstrates that our randomized policy consistently improves the consistency metric over both Purohit et al. (2018)-style baselines. Significant gains appear for the uniform families (unif100, unif200), the Gaussian (gauss) near $b$, and the geometric distribution (geom). The largest improvement occurs for the bi-modal twopoint distribution, where our method substantially outperforms both baselines.

### 6.2. Performance Comparison Under Prediction Error

**Setup.** We evaluate robustness to prediction error by perturbing the predicted distribution. Fix $(b, R) = (50, 1.7)$ and let the true skiing-day distribution be the discretized Gaussian $\mathcal{N}(90, 12^2)$ truncated to $\{1, \ldots, M\}$ (with $M$ chosen large enough to include essentially all mass). For each Wasserstein budget $\eta \geq 0$, we generate a perturbed prediction $\widehat{p}$ from $p$ via randomized mass transport with total transport cost at most $\eta$. We then compute the randomized buying-day distribution of each method using $\widehat{p}$ as input, but evaluate performance using the *true* distribution $p$.

**Evaluation Metric.** For each method, we report the (true) consistency $\text{Cons}(p) = \mathbb{E}[g_p(Z)]/\min_t g_p(t)$, where $g_p(\cdot)$

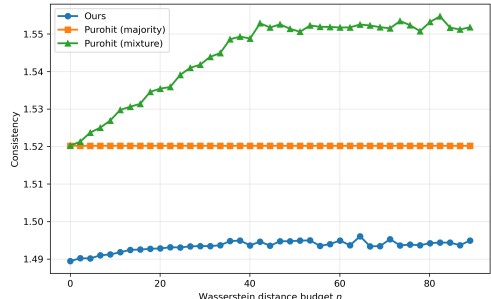

*Figure 1.* Consistency under prediction error. The x-axis is the Wasserstein perturbation budget $\eta$ used to generate $\widehat{p}$ from the true $p$; policies are computed from $\widehat{p}$, but the consistency is evaluated under $p$. The plot reports the mean over 25 random transports per $\eta$.

is defined from $p$ as in (5) and the expectation is taken over the actual PMF of $Z$ produced by the method (computed from $\widehat{p}$).

**Results.** Figure 1 shows that our policy remains consistently better than both Purohit et al. (2018)-style baselines across the full range of $\eta$. The majority-branch baseline stays essentially flat, while the mixture baseline degrades as $\eta$ increases, reflecting its sensitivity to distributional perturbations. In contrast, our method exhibits only mild degradation and maintains a stable advantage throughout.

## 7. Conclusion

We presented a unified framework for the ski rental problem that integrates distributional predictions while preserving rigorous robustness guarantees. Our approach bridges classical worst-case analysis and learning-augmented methods, enabling both deterministic and randomized algorithms to adapt smoothly to predictive accuracy. The proposed Clamp Policy and Water-Filling Algorithm achieve tight, interpretable competitive guarantees that recover classical bounds while improving consistency under predictions.

## Acknowledgement

This work was supported by the National Research Foundation of Korea (NRF) grant funded by the Korea government (MSIT) (No. RS-2026-25484051) and by the New Faculty Startup Fund of Seoul National University.

## Impact Statement

This work advances the theoretical foundations of online decision-making by providing robust methods for integrating distributional predictions into algorithm design. The proposed framework is primarily theoretical in nature and

does not introduce direct societal risks. By enabling more principled use of predictive information while maintaining worst-case guarantees, this research may contribute to more reliable decision-making systems in applications such as resource allocation and operations management.

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

# A. Proofs of Theoretical Results in Section 3

## A.1. Proof of Theorem 3.1 (Competitive Ratio for $t^* \leq b$)

*Proof.* The expected cost for a threshold $t^*$ is given by

$$g_p(t^*) = \sum_{d<t^*} p(d)d + \sum_{d \geq t^*} p(d)(b + t^* - 1).$$

The offline optimum is $\text{OPT}(p) = \sum_{d<b} p(d)d + b \cdot S(b)$, where $S(t) := \sum_{d \geq t} p(d)$. For the case where $t^* \leq b$, the difference between the expected cost and the optimum is:

$$g_p(t^*) - \text{OPT}(p) = \sum_{d=t^*}^{b-1} (b + t^* - 1 - d)p(d) + (t^* - 1)S(b).$$

To find the upper bound, we observe that for $d \in [t^*, b-1]$, the term $(b + t^* - 1 - d)$ is at most $(b-1)$. Additionally, we lower bound the denominator:

$$\text{OPT}(p) \geq \sum_{d=t^*}^{b-1} p(d)d + b \cdot S(b) \geq t^*(S(t^*) - S(b)) + b \cdot S(b).$$

Substituting these into the competitive ratio $\text{ECR}(p) = 1 + \frac{f_p(t^*) - \text{OPT}(p)}{\text{OPT}(p)}$:

$$\text{ECR}(p) \leq 1 + \frac{(b-1)(S(t^*) - S(b)) + (t^* - 1)S(b)}{t^*(S(t^*) - S(b)) + b \cdot S(b)}.$$

By dividing the numerator and denominator by $S(b)$ and letting $r = \frac{S(t^*)}{S(b)}$, we obtain:

$$\text{ECR}(p) \leq 1 + \frac{(b-1)(r-1) + (t^* - 1)}{t^*(r-1) + b} = 1 + \frac{(b-1)r - (b - t^*)}{t^*r + (b - t^*)}.$$

□

## A.2. Proof of Theorem 3.2 (Competitive Ratio for $t^* > b$)

*Proof.* When $t^* \geq b + 1$, the difference between the expected cost and the offline optimum is:

$$g_p(t^*) - \text{OPT}(p) = \sum_{d=b}^{t^*-1} (d - b)p(d) + (t^* - 1)S(t^*).$$

For any $d$ in the range $[b, t^* - 1]$, we have $d - b \leq t^* - 1 - b$. Thus, the summation is bounded as follows:

$$\sum_{d=b}^{t^*-1} (d - b)p(d) \leq (t^* - 1 - b)(S(b) - S(t^*)).$$

Using the inequality $\text{OPT}(p) \geq b \cdot S(b)$:

$$\text{ECR}(p) \leq 1 + \frac{(t^* - 1 - b)(S(b) - S(t^*)) + (t^* - 1)S(t^*)}{bS(b)} \leq 1 + \frac{(t^* - 1 - b)S(b) + bS(t^*)}{bS(b)}.$$

Letting $r = \frac{S(t^*)}{S(b)}$, we arrive at the simplified bound:

$$\text{ECR}(p) \leq \frac{t^* - 1}{b} + r.$$

□

### A.3. Proof of Theorem 3.3

*Proof.* Fix $C > 1$ and write $t^* = \alpha b$. Define $r := \frac{S(t^*)}{S(b)}$.

**Case 1: Early Buy ($t^* \le b$, i.e., $\alpha \le 1$).** From Theorem 3.1, we have

$$\mathrm{ECR}(p) \le 1 + \frac{(b-1)r - (b-t^*)}{t^* r + (b-t^*)}.$$

It suffices to require

$$\frac{(b-1)r - (b-t^*)}{t^* r + (b-t^*)} \le C - 1.$$

Equivalently,

$$\big((b-1) - (C-1)t^*\big)r \le C(b-t^*).$$

Whenever $(b-1) - (C-1)t^* > 0$, this implies

$$r \le \frac{C(b-t^*)}{(b-1) - (C-1)t^*}.$$

Substituting $t^* = \alpha b$ and dividing numerator and denominator by $b$ yields

$$r(\alpha) = \frac{S(\alpha b)}{S(b)} \le \frac{C(1-\alpha)}{1 - \frac{1}{b} - (C-1)\alpha}, \qquad \text{where } 1 - \frac{1}{b} - (C-1)\alpha > 0.$$

**Case 2: Late Buy ($t^* > b$, i.e., $\alpha \ge 1$).** From Theorem 3.2 and $r = \frac{S(t^*)}{S(b)}$, we have

$$\mathrm{ECR}(p) \le \frac{t^* - 1}{b} + r.$$

Thus, a sufficient condition for $\mathrm{ECR}(p) \le C$ is

$$\frac{t^* - 1}{b} + r \le C \iff r \le C - \frac{t^* - 1}{b} = C - \alpha + \frac{1}{b},$$

together with $C - \alpha + \frac{1}{b} \ge 0$.

This completes the proof. $\qquad\square$

## B. Proofs of Theoretical Results in Section 4

### B.1. Proofs of Stability and Distribution-Free Bounds

*Proof of Lemma 4.2.* By Kantorovich–Rubinstein duality,

$$W_1(p, \hat{p}) = \sup_{\|h\|_{\mathrm{Lip}} \le 1} \left| \mathbb{E}_{D \sim p}[h(D)] - \mathbb{E}_{D \sim \hat{p}}[h(D)] \right|.$$

Fix $t \in \mathbb{N}$. The function $d \mapsto \mathrm{cost}_{\mathcal{A}_t}(d)$ satisfies

$$\left| \mathrm{cost}_{\mathcal{A}_t}(d+1) - \mathrm{cost}_{\mathcal{A}_t}(d) \right| \le b \qquad \text{for every } d \in \mathbb{N}.$$

Indeed, this difference is 1 before day $t-1$, equals $b$ when crossing from $t-1$ to $t$, and is 0 thereafter. Hence $d \mapsto \mathrm{cost}_{\mathcal{A}_t}(d)$ is $b$-Lipschitz, and therefore

$$\left| \mathbb{E}_{D \sim p}\left[\mathrm{cost}_{\mathcal{A}_t}(D)\right] - \mathbb{E}_{D \sim \hat{p}}\left[\mathrm{cost}_{\mathcal{A}_t}(D)\right] \right| \le b W_1(p, \hat{p}) = b\eta.$$

In particular,

$$\mathbb{E}_{D\sim p}\left[\mathrm{cost}_{\mathcal{A}_t}(D)\right] \leq \mathbb{E}_{D\sim\hat{p}}\left[\mathrm{cost}_{\mathcal{A}_t}(D)\right] + b\eta.$$

Similarly, $d \mapsto \mathrm{OPT}(d) = \min\{d, b\}$ is 1-Lipschitz. Thus

$$\left|\mathbb{E}_{D\sim p}[\mathrm{OPT}(D)] - \mathbb{E}_{D\sim\hat{p}}[\mathrm{OPT}(D)]\right| \leq W_1(p, \hat{p}) = \eta,$$

which implies

$$\mathbb{E}_{D\sim p}[\mathrm{OPT}(D)] \geq \mathbb{E}_{D\sim\hat{p}}[\mathrm{OPT}(D)] - \eta.$$

$\square$

*Proof of Lemma 4.3.* We first bound the pointwise competitive ratio of $\mathcal{A}_t$.

Suppose first that $t \leq b$. If $d < t$, then

$$\mathrm{cost}_{\mathcal{A}_t}(d) = d = \mathrm{OPT}(d).$$

If $t \leq d < b$, then

$$\frac{\mathrm{cost}_{\mathcal{A}_t}(d)}{\mathrm{OPT}(d)} = \frac{b+t-1}{d} \leq \frac{b+t-1}{t} = 1 + \frac{b-1}{t}.$$

If $d \geq b$, then

$$\frac{\mathrm{cost}_{\mathcal{A}_t}(d)}{\mathrm{OPT}(d)} = \frac{b+t-1}{b} \leq \frac{b+t-1}{t} = 1 + \frac{b-1}{t},$$

where the inequality follows from $t \leq b$. Hence

$$\mathrm{WCR}_{\mathcal{A}_t} \leq 1 + \frac{b-1}{t}.$$

Now suppose that $t \geq b$. If $d < t$, then

$$\frac{\mathrm{cost}_{\mathcal{A}_t}(d)}{\mathrm{OPT}(d)} = \frac{d}{\min\{d, b\}} \leq \max\left\{1, \frac{t-1}{b}\right\} \leq 1 + \frac{t-1}{b}.$$

If $d \geq t$, then

$$\frac{\mathrm{cost}_{\mathcal{A}_t}(d)}{\mathrm{OPT}(d)} = \frac{b+t-1}{b} = 1 + \frac{t-1}{b}.$$

Therefore,

$$\mathrm{WCR}_{\mathcal{A}_t} \leq 1 + \frac{t-1}{b}.$$

Finally, for any distribution $q$,

$$\begin{aligned}
\mathrm{ECR}_{\mathcal{A}_t}(q) &= \frac{\mathbb{E}_{D\sim q}\left[\mathrm{cost}_{\mathcal{A}_t}(D)\right]}{\mathbb{E}_{D\sim q}[\mathrm{OPT}(D)]}\\
&= \sum_{d\in\mathbb{N}} \frac{q(d)\mathrm{OPT}(d)}{\mathbb{E}_{D\sim q}[\mathrm{OPT}(D)]} \frac{\mathrm{cost}_{\mathcal{A}_t}(d)}{\mathrm{OPT}(d)}.
\end{aligned}$$

The coefficients in this sum are nonnegative and sum to one. Thus $\mathrm{ECR}_{\mathcal{A}_t}(q)$ is a weighted average of the pointwise competitive ratios and is bounded by $\mathrm{WCR}_{\mathcal{A}_t}$. $\square$

### B.2. Proof of Theorem 4.4

*Proof.* The Clamp Policy is the threshold policy $\mathcal{A}_{\tilde{t}}$. By its definition,

$$\lceil \lambda b \rceil \le \tilde{t} \le \lfloor b/\lambda \rfloor,$$

and hence

$$\lambda b \le \tilde{t} \le \frac{b}{\lambda}.$$

If $\tilde{t} \le b$, Lemma 4.3 gives

$$
\begin{aligned}
\mathrm{WCR}_{\mathcal{A}_{\mathrm{Clamp}}} &= \mathrm{WCR}_{\mathcal{A}_{\tilde{t}}} \\
&\le 1 + \frac{b-1}{\tilde{t}} \\
&\le 1 + \frac{b-1}{\lambda b} \\
&= 1 + \frac{1}{\lambda} - \frac{1}{\lambda b} \\
&\le 1 + \frac{1}{\lambda} - \frac{1}{b}.
\end{aligned}
$$

If $\tilde{t} \ge b$, the same lemma gives

$$
\begin{aligned}
\mathrm{WCR}_{\mathcal{A}_{\mathrm{Clamp}}} &= \mathrm{WCR}_{\mathcal{A}_{\tilde{t}}} \\
&\le 1 + \frac{\tilde{t}-1}{b} \\
&\le 1 + \frac{b/\lambda - 1}{b} \\
&= 1 + \frac{1}{\lambda} - \frac{1}{b}.
\end{aligned}
$$

Therefore,

$$\mathrm{WCR}_{\mathcal{A}_{\mathrm{Clamp}}} \le 1 + \frac{1}{\lambda} - \frac{1}{b}.$$

Since the expected competitive ratio is bounded by the worst-case competitive ratio, this also implies

$$\mathrm{ECR}_{\mathcal{A}_{\mathrm{Clamp}}}(p) \le 1 + \frac{1}{\lambda} - \frac{1}{b}.$$

For the prediction-dependent bound, Lemma 4.2 gives

$$\mathbb{E}_{D \sim p}\left[\mathsf{cost}_{\mathcal{A}_{\tilde{t}}}(D)\right] \le \mathbb{E}_{D \sim \hat{p}}\left[\mathsf{cost}_{\mathcal{A}_{\tilde{t}}}(D)\right] + b\eta$$

and

$$\mathbb{E}_{D \sim p}[\mathrm{OPT}(D)] \ge \mathbb{E}_{D \sim \hat{p}}[\mathrm{OPT}(D)] - \eta.$$

Since

$$\theta = \frac{\eta}{\mathbb{E}_{D \sim \hat{p}}[\mathrm{OPT}(D)]} < 1,$$

the denominator below is positive, and hence

$$
\begin{aligned}
\mathrm{ECR}_{\mathcal{A}_{\mathrm{Clamp}}}(p) &= \frac{\mathbb{E}_{D \sim p}\left[\mathsf{cost}_{\mathcal{A}_{\tilde{t}}}(D)\right]}{\mathbb{E}_{D \sim p}[\mathrm{OPT}(D)]} \\
&\le \frac{\mathbb{E}_{D \sim \hat{p}}\left[\mathsf{cost}_{\mathcal{A}_{\tilde{t}}}(D)\right] + b\eta}{\mathbb{E}_{D \sim \hat{p}}[\mathrm{OPT}(D)] - \eta} \\
&= \frac{\mathrm{ECR}_{\mathcal{A}_{\tilde{t}}}(\hat{p}) + b\theta}{1 - \theta}.
\end{aligned}
$$

Combining the two bounds proves (1). □

## B.3. Proof of Corollary 4.5

*Proof.* The robustness statement follows immediately from Theorem 4.4:

$$\mathrm{WCR}_{\mathcal{A}_{\mathrm{Clamp}}} \leq 1 + \frac{1}{\lambda} - \frac{1}{b}.$$

For consistency, fix the predicted distribution $\hat{p}$ and the corresponding clamped threshold $\tilde{t}$. The proof of Lemma 4.2 gives the two-sided estimates

$$\left| \mathbb{E}_{D \sim p} \left[ \mathsf{cost}_{\mathcal{A}_{\tilde{t}}}(D) \right] - \mathbb{E}_{D \sim \hat{p}} \left[ \mathsf{cost}_{\mathcal{A}_{\tilde{t}}}(D) \right] \right| \leq b\eta$$

and

$$\left| \mathbb{E}_{D \sim p}[\mathrm{OPT}(D)] - \mathbb{E}_{D \sim \hat{p}}[\mathrm{OPT}(D)] \right| \leq \eta.$$

Consequently, as $\eta = W_1(p, \hat{p}) \to 0$, both the numerator and denominator defining $\mathrm{ECR}_{\mathcal{A}_{\tilde{t}}}(p)$ converge to their counterparts under $\hat{p}$. Since

$$\mathbb{E}_{D \sim \hat{p}}[\mathrm{OPT}(D)] > 0,$$

we obtain

$$\mathrm{ECR}_{\mathcal{A}_{\mathrm{Clamp}}}(p) = \mathrm{ECR}_{\mathcal{A}_{\tilde{t}}}(p) \longrightarrow \mathrm{ECR}_{\mathcal{A}_{\tilde{t}}}(\hat{p}).$$

In particular, if $p = \hat{p}$, then

$$\mathrm{ECR}_{\mathcal{A}_{\mathrm{Clamp}}}(\hat{p}) = \mathrm{ECR}_{\mathcal{A}_{\tilde{t}}}(\hat{p}).$$

$\square$

## B.4. Proof of the Total Variation Bound

Let

$$\eta_{\mathrm{TV}} := \frac{1}{2}\|p - \hat{p}\|_1 = \sup_{A \subseteq \mathbb{N}} |p(A) - \hat{p}(A)|.$$

For any bounded function $\varphi : \mathbb{N} \to \mathbb{R}$,

$$\left| \mathbb{E}_{D \sim p}[\varphi(D)] - \mathbb{E}_{D \sim \hat{p}}[\varphi(D)] \right| \leq \eta_{\mathrm{TV}} \left( \sup_d \varphi(d) - \inf_d \varphi(d) \right). \tag{7}$$

Indeed, writing

$$\varphi(d) = m + (M - m)\psi(d), \qquad m := \inf_d \varphi(d), \qquad M := \sup_d \varphi(d),$$

for some $\psi : \mathbb{N} \to [0, 1]$, we obtain

$$\left| \mathbb{E}_{D \sim p}[\psi(D)] - \mathbb{E}_{D \sim \hat{p}}[\psi(D)] \right| \leq \eta_{\mathrm{TV}},$$

which proves (7).

For any fixed $t \in \mathbb{N}$,

$$\sup_d \mathsf{cost}_{\mathcal{A}_t}(d) - \inf_d \mathsf{cost}_{\mathcal{A}_t}(d) \leq b + t - 1.$$

Applying (7) therefore gives

$$\mathbb{E}_{D \sim p} \left[ \mathsf{cost}_{\mathcal{A}_t}(D) \right] \leq \mathbb{E}_{D \sim \hat{p}} \left[ \mathsf{cost}_{\mathcal{A}_t}(D) \right] + \eta_{\mathrm{TV}}(b + t - 1). \tag{8}$$

Moreover, since $0 \leq \mathrm{OPT}(d) \leq b$,

$$\mathbb{E}_{D \sim p}[\mathrm{OPT}(D)] \geq \mathbb{E}_{D \sim \hat{p}}[\mathrm{OPT}(D)] - b\eta_{\mathrm{TV}}. \tag{9}$$

Applying (8) and (9) with $t = \tilde{t}$, and assuming

$$\mathbb{E}_{D \sim \hat{p}}[\mathrm{OPT}(D)] > b\eta_{\mathrm{TV}},$$

we obtain

$$\begin{aligned}
\text{ECR}_{\mathcal{A}_{\text{Clamp}}}(p) &= \frac{\mathbb{E}_{D\sim p}\left[\text{cost}_{\mathcal{A}_{\tilde{t}}}(D)\right]}{\mathbb{E}_{D\sim p}[\text{OPT}(D)]} \\
&\leq \frac{\mathbb{E}_{D\sim\hat{p}}\left[\text{cost}_{\mathcal{A}_{\tilde{t}}}(D)\right] + \eta_{\text{TV}}(b+\tilde{t}-1)}{\mathbb{E}_{D\sim\hat{p}}[\text{OPT}(D)] - b\eta_{\text{TV}}} \\
&= \frac{\text{ECR}_{\mathcal{A}_{\tilde{t}}}(\hat{p}) + \dfrac{\eta_{\text{TV}}(b+\tilde{t}-1)}{\mathbb{E}_{D\sim\hat{p}}[\text{OPT}(D)]}}{1 - \dfrac{b\eta_{\text{TV}}}{\mathbb{E}_{D\sim\hat{p}}[\text{OPT}(D)]}}.
\end{aligned}$$

Finally, the worst-case guarantee

$$\text{WCR}_{\mathcal{A}_{\text{Clamp}}} \leq 1 + \frac{1}{\lambda} - \frac{1}{b}$$

depends only on the range of $\tilde{t}$ and the distribution-free threshold bound. It is therefore unchanged when prediction error is measured by total variation distance.

## C. Proofs of Theoretical Results in Section 5

### C.1. Proof of Lemma 5.1 (Robustness Constraints)

*Proof.* Recall

$$F(x) = \sum_{t=1}^{x} f(t), \qquad \mu(x) = \sum_{t=1}^{x}(t-1)f(t), \qquad \mu(\infty) = \sum_{t=1}^{\infty}(t-1)f(t).$$

A direct expansion yields

$$\mathbb{E}_{z\sim f}[C_z(x)] = \sum_{z\leq x} f(z)(b+z-1) + \sum_{z>x} f(z)\,x = bF(x) + \mu(x) + x\big(1-F(x)\big) = \mu(x) + (b-x)F(x) + x.$$

($\Rightarrow$) **Necessity.** Assume $R$-robustness: $\mathbb{E}[C_Z(x)] \leq R\,\text{OPT}(x)$ for all $x$.

If $1 \leq x \leq b-1$, then $\text{OPT}(x) = x$, hence

$$\mu(x) + (b-x)F(x) + x \leq Rx \iff \mu(x) + (b-x)F(x) \leq (R-1)x.$$

For $x \geq b$, robustness gives $\mathbb{E}[C_Z(x)] \leq Rb$ for all $x \geq b$. Define $g(x) := \mathbb{E}[C_Z(x)] = bF(x) + \mu(x) + x(1-F(x))$. Since $F(u) \to 1$ and $\mu(u) \to \mu(\infty)$, we have $\lim_{u\to\infty} g(u) = b + \mu(\infty)$. Thus $b + \mu(\infty) \leq Rb$, i.e., $\mu(\infty) \leq (R-1)b$.

($\Leftarrow$) **Sufficiency.** If $1 \leq x \leq b-1$, we get

$$\mathbb{E}[C_Z(x)] = \mu(x) + (b-x)F(x) + x \leq (R-1)x + x = Rx = R\,\text{OPT}(x).$$

For $x \geq b$, Using $F(x+1) = F(x) + f(x+1)$ and $\mu(x+1) = \mu(x) + xf(x+1)$, we obtain

$$g(x+1) - g(x) = (1-F(x)) + (b-1)f(x+1) \geq 0,$$

so $g(x)$ is nondecreasing in $x$ and therefore $g(x) \leq \lim_{u\to\infty} g(u)$. Therefore,

$$\mathbb{E}[C_Z(x)] = g(x) \leq \lim_{u\to\infty} g(u) = b + \mu(\infty) \leq b + (R-1)b = Rb = R\,\text{OPT}(x).$$

Hence, the policy is $R$-robust. $\qquad\square$

## C.2. Proof of Theorem 5.2

*Proof.* We minimize the linear objective $\sum_{z \geq 1} g(z) f(z)$ subject to the robustness constraints

$$H(x) := \mu(x) + (b - x)F(x) \leq (R - 1)x \quad (1 \leq x \leq b - 1), \qquad \mu(\infty) \leq (R - 1)b,$$

where $F(x) = \sum_{t=1}^{x} f(t)$ and $\mu(x) = \sum_{t=1}^{x}(t - 1)f(t)$.

Among all optimal solutions, fix one that minimizes $\mu(\infty)$.

**Tightness on the support (for $x \leq b - 1$).** We claim that for this chosen optimal solution, $H(x) = (R - 1)x$ holds for every $x \leq b - 1$ such that $f(x + 1) > 0$. Suppose, for contradiction, that there exists $x_0 < b$ with $f(x_0 + 1) > 0$ and strict slack

$$H(x_0) < (R - 1)x_0.$$

Since $g$ is monotone increasing, we can shift a small mass from $x_0 + 1$ to $x_0$ without increasing the objective. Fix $\varepsilon \in (0, f(x_0 + 1)]$ and define a new distribution $f_1$ by

$$f_1(x_0) = f(x_0) + \varepsilon, \qquad f_1(x_0 + 1) = f(x_0 + 1) - \varepsilon, \qquad f_1(t) = f(t) \ (t \notin \{x_0, x_0 + 1\}).$$

Let $F_1, \mu_1, H_1$ denote the corresponding quantities under $f_1$. We verify feasibility.

*Tail constraint ($x \geq b$).* Only the terms at $t = x_0$ and $t = x_0 + 1$ change, so

$$\mu_1(\infty) - \mu(\infty) = (x_0 - 1)\varepsilon - x_0\varepsilon = -\varepsilon < 0.$$

Hence $\mu_1(\infty) \leq \mu(\infty) \leq (R - 1)b$.

*Pointwise constraints ($x \leq b - 1$).* For $x < x_0$, both $F(x)$ and $\mu(x)$ are unchanged. For $x \geq x_0 + 1$, the CDF is unchanged (mass is moved internally), i.e., $F_1(x) = F(x)$, while

$$\mu_1(x) - \mu(x) = (x_0 - 1)\varepsilon - x_0\varepsilon = -\varepsilon,$$

so $H_1(x) = H(x) - \varepsilon$ becomes strictly smaller and constraints remain satisfied. It remains to check $x = x_0$. Here $F_1(x_0) = F(x_0) + \varepsilon$ and $\mu_1(x_0) = \mu(x_0) + (x_0 - 1)\varepsilon$, hence

$$H_1(x_0) - H(x_0) = \big(\mu_1(x_0) - \mu(x_0)\big) + (b - x_0)\big(F_1(x_0) - F(x_0)\big) = (x_0 - 1)\varepsilon + (b - x_0)\varepsilon = (b - 1)\varepsilon.$$

Because $(R - 1)x_0 - H(x_0) > 0$, choosing

$$0 < \varepsilon \leq \min\left\{ f(x_0 + 1), \ \frac{(R - 1)x_0 - H(x_0)}{b - 1} \right\}$$

ensures $H_1(x_0) \leq (R - 1)x_0$. Thus $f_1$ is feasible.

Finally, the objective does not increase:

$$\sum_z g(z)f_1(z) - \sum_z g(z)f(z) = \varepsilon\big(g(x_0) - g(x_0 + 1)\big) \leq 0.$$

Hence $f_1$ is also optimal. If the above inequality is strict, this contradicts the optimality of $f$. If equality holds, then $\mu_1(\infty) = \mu(\infty) - \varepsilon < \mu(\infty)$, contradicting our choice of $f$ as an optimal solution minimizing $\mu(\infty)$. Therefore, for the chosen optimal distribution,

$$\mu(x) + (b - x)F(x) = (R - 1)x \quad \text{for all } 1 \leq x \leq b - 1 \text{ such that } f(x + 1) > 0.$$

**Linear recurrence on the tight region.** On indices where the constraint is tight at both $x$ and $x + 1$,

$$\mu(x + 1) + (b - x - 1)F(x + 1) - \big(\mu(x) + (b - x)F(x)\big) = R - 1.$$

Using $\mu(x + 1) - \mu(x) = x f(x + 1)$ and $F(x + 1) - F(x) = f(x + 1)$, we obtain

$$x f(x + 1) + (b - x - 1)\big(F(x) + f(x + 1)\big) - (b - x)F(x) = R - 1,$$

which simplifies to

$$(b - 1)f(x + 1) - F(x) = R - 1.$$

Equivalently,

$$F(x + 1) = F(x) + f(x + 1) = \left(1 + \frac{1}{b - 1}\right) F(x) + \frac{R - 1}{b - 1}.$$

**Geometric solution and truncation.**   With $F(0) = 0$, the recurrence yields

$$F^*(x) = (R - 1)\left[\left(1 + \frac{1}{b - 1}\right)^x - 1\right]$$

until $F^*(x)$ reaches 1. Since $F$ must be a CDF, we truncate at the smallest $x_0$ such that $F^*(x_0) \geq 1$ and place the remaining mass at $x_0$ so that $F(x) = 1$ for all $x \geq x_0$. □

## C.3. Proof of Theorem 5.3

*Proof.* Let $F^\star$ denote the geometric CDF from Theorem 5.2, and let $f^\star$ be its corresponding pmf:

$$f^\star(t) := F^\star(t) - F^\star(t - 1), \qquad F^\star(0) := 0.$$

Write

$$\gamma := 1 + \frac{1}{b - 1} = \frac{b}{b - 1}, \qquad F^\star(x) = \min\big((R - 1)(\gamma^x - 1),\, 1\big).$$

Let

$$x_0 := \min\{x \in \mathbb{N} : F^\star(x) = 1\}.$$

Equivalently,

$$x_0 = \min\{x \in \mathbb{N} : (R - 1)(\gamma^x - 1) \geq 1\} = \left\lceil \frac{\ln\big(R/(R - 1)\big)}{\ln\big(b/(b - 1)\big)} \right\rceil.$$

Thus $f^\star$ is supported on $\{1, \ldots, x_0\}$.

By assumption,

$$y \geq b - 1 + \frac{\ln(R/(R - 1))}{\ln(b/(b - 1))}.$$

Since $y + 1 - b$ is an integer, this implies

$$y + 1 - b \geq \left\lceil \frac{\ln(R/(R - 1))}{\ln(b/(b - 1))} \right\rceil = x_0.$$

Hence

$$x_0 \leq y + 1 - b. \tag{10}$$

We next show that every index strictly larger than $x_0$ has cost at least $g(x_0)$. If $x_0 < t \leq y$, then by the first condition of Theorem 5.3,

$$g(t) \geq g(x_0).$$

If $t \geq y + 1$, then by the second condition of Theorem 5.3 and (10),

$$g(t) \geq g(y + 1 - b) \geq g(x_0),$$

where the last inequality again follows from the first condition of Theorem 5.3. Therefore,

$$g(t) \geq g(x_0) \qquad \text{for all } t \geq x_0 + 1. \tag{11}$$

Now define an auxiliary cost function

$$\widetilde{g}(t) := \begin{cases} g(t), & 1 \leq t \leq x_0, \\ g(x_0), & t \geq x_0 + 1. \end{cases}$$

By (11), we have

$$\widetilde{g}(t) \leq g(t) \qquad \forall t \in \mathbb{N}. \tag{12}$$

Moreover,

$$\widetilde{g}(1) \leq \widetilde{g}(2) \leq \cdots,$$

because $x_0 \leq y$ and the first condition of Theorem 5.3 implies

$$g(1) \leq g(2) \leq \cdots \leq g(x_0),$$

while $\widetilde{g}$ is constant on $\{x_0 + 1, x_0 + 2, \dots\}$. Therefore, Theorem 5.2 implies that $f^\star$ is optimal for

$$\sum_{t \geq 1} \widetilde{g}(t) f(t)$$

over all $R$-robust stopping distributions $f$. Hence,

$$\sum_{t \geq 1} \widetilde{g}(t) f(t) \geq \sum_{t \geq 1} \widetilde{g}(t) f^\star(t) \qquad \text{for every } R\text{-robust } f. \tag{13}$$

Since $f^\star$ is supported on $\{1, \dots, x_0\}$ and $\widetilde{g}(t) = g(t)$ for $t \leq x_0$, we have

$$\sum_{t \geq 1} \widetilde{g}(t) f^\star(t) = \sum_{t \geq 1} g(t) f^\star(t). \tag{14}$$

We now argue by contradiction. Assume that $F^\star$ is not optimal for the original objective $g$. Then there exists an $R$-robust stopping distribution $f$ such that

$$\sum_{t \geq 1} g(t) f(t) < \sum_{t \geq 1} g(t) f^\star(t).$$

But by (12), (13), and (14),

$$\sum_{t \geq 1} g(t) f(t) \geq \sum_{t \geq 1} \widetilde{g}(t) f(t) \geq \sum_{t \geq 1} \widetilde{g}(t) f^\star(t) = \sum_{t \geq 1} g(t) f^\star(t),$$

which is a contradiction. Hence $F^\star$ is optimal for the original objective $g$ as well.

Finally, consider the one-hot prediction case $p(y) = 1$. Then

$$g(t) = \begin{cases} b + t - 1, & t \leq y, \\ y, & t \geq y + 1. \end{cases}$$

Thus the first condition of Theorem 5.3 holds trivially. Also, for every $t \geq y + 1$,

$$g(t) = y = b + (y + 1 - b) - 1 = g(y + 1 - b),$$

so the second condition of Theorem 5.3 holds as well. Therefore, the theorem applies whenever

$$y \geq b - 1 + \frac{\ln(R/(R-1))}{\ln(b/(b-1))}.$$

$\square$

### C.4. Exact solution for the one-hot objective for all $y$

In this subsection we fix integers $b > 1$ and $y \geq 1$, and consider the one-hot objective

$$g(t) = \begin{cases} b + t - 1, & 1 \leq t \leq y, \\ y, & t \geq y + 1. \end{cases}$$

We study the linear program

$$\min_f \sum_{t=1}^{\infty} g(t) f(t) \qquad \text{s.t. } f \text{ satisfies the robustness constraints in Lemma 5.1.}$$

Write

$$F(x) := \sum_{t=1}^{x} f(t), \qquad \mu(x) := \sum_{t=1}^{x} (t-1) f(t), \qquad H(x) := \mu(x) + (b - x) F(x).$$

Then

$$H(x) = b F(x) - \sum_{u=1}^{x} F(u).$$

Also define

$$\gamma := \frac{b}{b-1}, \qquad G(x) := (R-1)(\gamma^x - 1), \qquad x \in \mathbb{Z}_{\geq 0}.$$

For each $p \in [0, 1]$, define the capped geometric envelope

$$F_p(x) := \min\{G(x), p\}, \qquad x \in \mathbb{Z}_{\geq 0}.$$

The next theorem gives an exact optimizer for every $y \geq 1$.

**Theorem C.1** (Exact optimizer for the one-hot objective for all $y$). *Assume that the feasible set of Lemma 5.1 is nonempty.*

*(i) **The case** $1 \leq y \leq b - 1$**. Define***

$$S_y(p) := \sum_{x=1}^{y} F_p(x),$$

$$\Lambda_y(p) := \gamma^{b-y-1} \frac{(R-1)(y+1) + S_y(p)}{b-1} + (R-1)(\gamma^{b-y-1} - 1),$$

*and let*

$$p^\star := \min\{p \in [0, 1] : \Lambda_y(p) \geq 1\}.$$

*Then*

$$F^\star(x) = \begin{cases} F_{p^\star}(x), & 0 \leq x \leq y, \\ \min\left\{\gamma^{x-y-1} \frac{(R-1)(y+1)+S_y(p^\star)}{b-1} + (R-1)(\gamma^{x-y-1} - 1), 1\right\}, & y+1 \leq x \leq b, \\ 1, & x \geq b, \end{cases}$$

*is an optimal CDF.*

*The minimum objective value is*

$$y + b p^\star - S_y(p^\star).$$

*(ii) **The case** $y \geq b$**. Define***

$$\Phi_y(p) := (y - b)p + \sum_{x=1}^{b} F_p(x), \qquad p \in [0, 1],$$

$$m := \min\{y - b + 1, b\}, \qquad p_{\text{cheap}} := \min\{G(m), 1\}, \qquad \Delta_y := \max\{y - (R-1)b, 0\},$$

*and let*

$$p^{\star} := \min\left\{p \in [0,1] : \Phi_y(p) \geq \max\{\Delta_y, \Phi_y(p_{\text{cheap}})\}\right\}.$$

*Then*

$$F^{\star}(x) = \begin{cases} F_{p^{\star}}(x), & 0 \leq x \leq b, \\ p^{\star}, & b+1 \leq x \leq y, \\ 1, & x \geq y+1, \end{cases}$$

*is an optimal CDF.*

*The minimum objective value is*

$$y + b\,p^{\star} - \Phi_y(p^{\star}).$$

*In both cases the corresponding pmf is*

$$f^{\star}(t) = F^{\star}(t) - F^{\star}(t-1), \qquad t \geq 1.$$

*Proof.* We divide the proof into several steps.

**Step 1: geometric envelope, nonemptiness implies $G(b) \geq 1$, and the front-loaded cap.** For every $1 \leq x \leq b-1$, the robustness constraint is

$$H(x) = bF(x) - \sum_{u=1}^{x} F(u) \leq (R-1)x.$$

Equivalently,

$$(b-1)F(x) \leq (R-1)x + \sum_{u=1}^{x-1} F(u). \tag{15}$$

On the other hand, the function $G(x) = (R-1)(\gamma^x - 1)$ satisfies

$$(b-1)G(x) = (R-1)x + \sum_{u=1}^{x-1} G(u) \qquad (x \geq 1).$$

Therefore an induction on $x$ using (15) gives

$$F(x) \leq G(x) \qquad (1 \leq x \leq b-1) \tag{16}$$

for every feasible $F$.

We next record a consequence of the assumption that the feasible set is nonempty. Take any feasible policy and move all probability mass on $\{t > b\}$ to the single day $b$. This does not affect the pointwise constraints for $x \leq b-1$, and it can only decrease $\mu(\infty)$. Hence there exists a feasible policy supported on $\{1, \ldots, b\}$. For such a policy, $F(b) = 1$. Since the support is contained in $\{1, \ldots, b\}$, we also have

$$\mu(\infty) = \mu(b).$$

Hence the tail constraint

$$\mu(\infty) \leq (R-1)b$$

becomes

$$\mu(b) \leq (R-1)b.$$

Now

$$H(b) = \mu(b) + (b-b)F(b) = \mu(b),$$

so this is equivalent to

$$H(b) \leq (R-1)b.$$

Using

$$H(b) = bF(b) - \sum_{u=1}^{b} F(u),$$

we obtain

$$bF(b) - \sum_{u=1}^{b} F(u) \le (R-1)b.$$

Separating the last term in the sum gives

$$bF(b) - \sum_{u=1}^{b-1} F(u) - F(b) \le (R-1)b,$$

that is,

$$(b-1)F(b) - \sum_{u=1}^{b-1} F(u) \le (R-1)b.$$

Therefore

$$(b-1)F(b) \le (R-1)b + \sum_{u=1}^{b-1} F(u).$$

Using (16) on the right-hand side gives

$$b - 1 \le (R-1)b + \sum_{u=1}^{b-1} G(u) = (b-1)G(b).$$

Thus

$$G(b) \ge 1. \tag{17}$$

In particular, for every $p \in [0,1]$ we have

$$F_p(b) = p. \tag{18}$$

Now fix $p \in [0,1]$. If a feasible $F$ satisfies $F(y) = p$, then trivially $F(x) \le p$ for $0 \le x \le y$. Combining this with (16) yields

$$F(x) \le F_p(x) := \min\{G(x), p\} \qquad (0 \le x \le y). \tag{19}$$

Likewise, if a feasible $F$ satisfies $F(b) = p$, then by (16) and (18),

$$F(x) \le F_p(x) \qquad (0 \le x \le b). \tag{20}$$

Finally, the capped envelope $F_p$ itself is feasible for the pointwise constraints $x \le b - 1$. The claim is trivial for $p = 0$, so assume $p > 0$ and let

$$k(p) := \min\{x \ge 0 : G(x) \ge p\}.$$

If $k(p) \ge b$, then $F_p(x) = G(x)$ for all $0 \le x \le b - 1$, so all pointwise constraints are tight there.

Assume now that $1 \le k(p) \le b - 1$. For $x < k(p)$ we still have $F_p(x) = G(x)$, so the constraints are tight there. At $x = k(p)$,

$$H_p(k(p)) = bF_p(k(p)) - \sum_{u=1}^{k(p)} F_p(u) = (b-1)p - \sum_{u=1}^{k(p)-1} G(u)$$

and since $p \le G(k(p))$,

$$H_p(k(p)) \le (b-1)G(k(p)) - \sum_{u=1}^{k(p)-1} G(u) = (R-1)k(p).$$

For every $x \ge k(p)$, the CDF is flat:

$$F_p(x+1) = F_p(x) = p.$$

Hence

$$H_p(x+1) - H_p(x) = b\big(F_p(x+1) - F_p(x)\big) - F_p(x+1) = -p \le 0.$$

Therefore $H_p(x) \leq H_p(k(p)) \leq (R-1)k(p) \leq (R-1)x$ for all $x \geq k(p)$. So $F_p$ is feasible for all pointwise constraints $x \leq b - 1$.

Thus $F_p$ is the canonical *front-loaded* feasible prefix with endpoint value $p$.

**Step 2: proof of part (i), i.e., $1 \leq y \leq b - 1$.**

Because $g(t) = y$ for every $t \geq y + 1$, moving any mass from a day $t > b$ to the day $b$ leaves the objective unchanged, does not affect the constraints $H(x) \leq (R-1)x$ for $x \leq b - 1$, and decreases $\mu(\infty)$. Hence, without loss of optimality, we may restrict to pmf's supported on $\{1, \ldots, b\}$. Then $F(b) = 1$, and the tail constraint becomes

$$\mu(b) \leq (R-1)b.$$

Using $H(b) = \mu(b) = bF(b) - \sum_{u=1}^{b} F(u)$, this is equivalently

$$(b-1)F(b) \leq (R-1)b + \sum_{u=1}^{b-1} F(u). \tag{21}$$

This is the same algebraic form as (15), but now it comes from the tail constraint after support compression.

Now fix a feasible policy and write

$$p := F(y).$$

Since $g(t) = y$ for $t \geq y + 1$ and the support is contained in $\{1, \ldots, b\}$,

$$\sum_{t=1}^{\infty} g(t)f(t) = \sum_{t=1}^{y} (b + t - 1)f(t) + y \sum_{t=y+1}^{b} f(t) = y + bF(y) - \sum_{x=1}^{y} F(x).$$

Therefore

$$\sum_{t=1}^{\infty} g(t)f(t) = y + b\,p - \sum_{x=1}^{y} F(x). \tag{22}$$

By (19),

$$\sum_{x=1}^{y} F(x) \leq \sum_{x=1}^{y} F_p(x) = S_y(p).$$

Hence every feasible policy with $F(y) = p$ has objective value at least

$$y + b\,p - S_y(p),$$

and equality is achieved exactly when the prefix on $\{0, 1, \ldots, y\}$ equals $F_p$.

So the remaining task is to determine for which $p$ this front-loaded prefix can be completed to a feasible CDF with $F(b) = 1$.

*Maximal continuation from day $y$.* Starting from the prefix $F_p$ on $\{0, \ldots, y\}$, define recursively

$$U_p(y) := p, \qquad U_p(x) := \frac{(R-1)x + \sum_{u=1}^{x-1} U_p(u)}{b-1}, \qquad y + 1 \leq x \leq b,$$

where the case $x = b$ uses (21). Because the right-hand side is monotone increasing in the earlier cumulative values, a straightforward induction on $x$ shows that every feasible continuation of the prefix $F_p$ satisfies

$$F(x) \leq U_p(x) \qquad (y + 1 \leq x \leq b).$$

Conversely, choosing equality in the recursion produces the maximal feasible continuation until the CDF reaches 1; after that one keeps the CDF equal to 1, which only makes all later constraints easier.

At $x = y + 1$,

$$U_p(y+1) = \frac{(R-1)(y+1) + S_y(p)}{b-1}.$$

For $x \geq y + 2$, subtracting the defining equalities for $U_p(x)$ and $U_p(x-1)$ gives

$$U_p(x) - U_p(x-1) = \frac{(R-1) + U_p(x-1)}{b-1},$$

that is,

$$U_p(x) = \gamma \, U_p(x-1) + \frac{R-1}{b-1}.$$

Solving this recurrence yields

$$U_p(x) = \gamma^{x-y-1} \frac{(R-1)(y+1) + S_y(p)}{b-1} + (R-1)(\gamma^{x-y-1} - 1), \qquad y+1 \leq x \leq b.$$

In particular,

$$\Lambda_y(p) = U_p(b) = \gamma^{b-y-1} \frac{(R-1)(y+1) + S_y(p)}{b-1} + (R-1)(\gamma^{b-y-1} - 1).$$

Therefore a prefix level $p$ is feasible if and only if $\Lambda_y(p) \geq 1$. Indeed: if a feasible continuation exists, then necessarily

$$1 = F(b) \leq U_p(b) = \Lambda_y(p).$$

Conversely, if $\Lambda_y(p) \geq 1$, then the maximal continuation reaches 1 by time $b$, and truncating at the first hitting time yields a feasible CDF with prefix $F_p$ and total mass 1.

*Why the smallest feasible $p$ is optimal.* Define

$$\Psi_y(p) := y + b\,p - S_y(p).$$

The function

$$S_y(p) = \sum_{x=1}^{y} \min\{G(x), p\}$$

is continuous, piecewise linear, and nondecreasing.

More precisely, if

$$G(k-1) \leq p \leq G(k) \qquad \text{for some } 1 \leq k \leq y,$$

then

$$S_y(p) = \sum_{x=1}^{k-1} G(x) + (y-k+1)p,$$

and hence

$$\Psi_y(p) = y - \sum_{x=1}^{k-1} G(x) + (b - y + k - 1)p.$$

Therefore, on each interval $[G(k-1), G(k)]$, the function $\Psi_y$ is affine with slope

$$b - y + k - 1 > 0.$$

If $G(y) \leq p \leq 1$, then $F_p(x) = G(x)$ for every $1 \leq x \leq y$, so

$$S_y(p) = \sum_{x=1}^{y} G(x),$$

and hence

$$\Psi_y(p) = y - \sum_{x=1}^{y} G(x) + bp,$$

which is affine with slope $b > 0$ on $[G(y), 1]$.

Since $\Psi_y$ is continuous and has strictly positive slope on every linearity interval, it is strictly increasing on $[0, 1]$.

Because the feasible set is nonempty, there exists at least one feasible policy supported on $\{1, \ldots, b\}$; for its value $p = F(y)$ we have just proved $\Lambda_y(p) \geq 1$. Hence the set

$$\{p \in [0, 1] : \Lambda_y(p) \geq 1\}$$

is nonempty. Since $\Lambda_y$ is continuous and nondecreasing, the number

$$p^\star := \min\{p \in [0, 1] : \Lambda_y(p) \geq 1\}$$

is well defined. Since $\Psi_y$ is strictly increasing, the minimum objective value is attained exactly at this smallest feasible prefix level $p^\star$, and the corresponding optimal CDF is the one stated in part (i).

**Step 3: proof of part (ii), i.e., $y \geq b$.**

We first compress the support.

*Compression of the far tail.* If $t \geq y + 2$, then $g(t) = g(y + 1) = y$. Moving any mass from $t$ to $y + 1$ therefore leaves the objective unchanged, does not affect the constraints $H(x) \leq (R - 1)x$ for $x \leq b - 1$, and decreases $\mu(\infty)$. So we may assume that there is no mass above $y + 1$.

*Compression of the middle block $\{b + 1, \ldots, y\}$.* Fix $t \in \{b + 1, \ldots, y\}$ and define

$$\alpha_t := \frac{y + 1 - t}{y - b + 1} \in [0, 1].$$

Then

$$\alpha_t(b - 1) + (1 - \alpha_t)y = t - 1.$$

Hence replacing an $\varepsilon$-mass at day $t$ by $\alpha_t \varepsilon$ at day $b$ and $(1 - \alpha_t)\varepsilon$ at day $y + 1$ preserves total mass and preserves the contribution to $\mu(\infty)$. Since all three days are $> b - 1$, this replacement does not affect any pointwise constraint for $x \leq b - 1$.

Its effect on the objective is

$$\varepsilon\big(\alpha_t g(b) + (1 - \alpha_t)g(y + 1) - g(t)\big)$$
$$= \varepsilon\big(\alpha_t(2b - 1) + (1 - \alpha_t)y - (b + t - 1)\big)$$
$$= -\varepsilon \frac{b(t - b)}{y - b + 1} \leq 0.$$

Therefore some optimal solution is supported on the set

$$\{1, 2, \ldots, b, y + 1\}.$$

Now fix such a feasible policy and let

$$p := F(b) = \sum_{t=1}^{b} f(t).$$

For this support pattern,

$$\mu(\infty) = \sum_{t=1}^{b}(t - 1)f(t) + y(1 - p).$$

Thus the tail constraint $\mu(\infty) \leq (R - 1)b$ is equivalent to

$$\sum_{t=1}^{b}(y + 1 - t)f(t) \geq y - (R - 1)b.$$

Since the left-hand side is nonnegative, this is equivalent to

$$\sum_{t=1}^{b}(y+1-t)f(t) \geq \Delta_y, \qquad \Delta_y := \max\{y-(R-1)b, 0\}.$$

Define the achieved reduction

$$\mathcal{R}(f) := \sum_{t=1}^{b}(y+1-t)f(t).$$

Using summation by parts,

$$\mathcal{R}(f) = (y-b)p + \sum_{x=1}^{b} F(x).$$

By (20),

$$\sum_{x=1}^{b} F(x) \leq \sum_{x=1}^{b} F_p(x),$$

so

$$\mathcal{R}(f) \leq \Phi_y(p), \qquad \Phi_y(p) := (y-b)p + \sum_{x=1}^{b} F_p(x).$$

Moreover, because of (18), the CDF $F_p$ indeed has endpoint value $F_p(b) = p$. Hence, whenever $\Phi_y(p) \geq \Delta_y$, equality is attained by taking the front-loaded cap $F_p$ on $\{0, 1, \ldots, b\}$, then keeping the CDF flat at level $p$ on $\{b+1, \ldots, y\}$, and finally placing the remaining mass $1-p$ at the single tail point $y+1$.

The objective can be written as

$$\sum_{t=1}^{\infty} g(t)f(t) = y + \sum_{t=1}^{b}(b-y+t-1)f(t) = y + b\,p - \mathcal{R}(f).$$

Therefore every feasible policy with early mass $p$ has objective value at least

$$y + b\,p - \Phi_y(p),$$

and equality is attained whenever $\Phi_y(p) \geq \Delta_y$ by the canonical construction just described.

So the problem reduces to the one-dimensional minimization

$$\min_{p \in [0,1]} \Big(y + b\,p - \Phi_y(p)\Big) \qquad \text{s.t. } \Phi_y(p) \geq \Delta_y.$$

*Shape of the one-dimensional objective.* Let

$$J_y(p) := y + b\,p - \Phi_y(p).$$

By Step 1, the nonemptiness of the feasible set implies $G(b) \geq 1$. Since $G(0) = 0$ and $G$ is increasing, every $p \in [0,1]$ belongs to some interval

$$G(k-1) \leq p \leq G(k) \qquad \text{with } 1 \leq k \leq b.$$

Fix such a $k$. Then

$$F_p(x) = \begin{cases} G(x), & 1 \leq x \leq k-1, \\ p, & k \leq x \leq b, \end{cases}$$

and therefore

$$\sum_{x=1}^{b} F_p(x) = \sum_{x=1}^{k-1} G(x) + (b-k+1)p.$$

Hence

$$\Phi_y(p) = (y - b)p + \sum_{x=1}^{k-1} G(x) + (b - k + 1)p = \sum_{x=1}^{k-1} G(x) + (y - k + 1)p,$$

so on the interval $[G(k-1), G(k)]$,

$$J_y(p) = y - \sum_{x=1}^{k-1} G(x) + \big(k - (y - b + 1)\big)p.$$

Thus $J_y$ is affine on each interval $[G(k-1), G(k)]$, with slope

$$k - (y - b + 1).$$

Now set

$$m := \min\{y - b + 1, b\}, \qquad p_{\text{cheap}} := \min\{G(m), 1\}.$$

Then:

- if $1 \le k \le m$, the slope $k - (y - b + 1) \le 0$, so $J_y$ is nonincreasing on $[G(k-1), G(k)]$;

- if $m + 1 \le k \le b$, the slope $k - (y - b + 1) \ge 0$, so $J_y$ is nondecreasing on $[G(k-1), G(k)]$.

Therefore $J_y$ is nonincreasing on $[0, p_{\text{cheap}}]$ and nondecreasing on $[p_{\text{cheap}}, 1]$. Consequently, among all feasible $p$ satisfying $\Phi_y(p) \ge \Delta_y$, the canonical minimizer is the smallest feasible $p$ lying at or to the right of $p_{\text{cheap}}$, equivalently the smallest $p$ such that

$$\Phi_y(p) \ge \max\{\Delta_y, \Phi_y(p_{\text{cheap}})\}.$$

That is,

$$p^\star = \min\Big\{p \in [0, 1] : \Phi_y(p) \ge \max\{\Delta_y, \Phi_y(p_{\text{cheap}})\}\Big\}.$$

The corresponding optimal CDF is exactly the one stated in part (ii).

This completes the proof. $\qquad\square$

*Remark* C.2 (Long-flat-tail regime and consistency with Theorem 5.3). Assume $y \ge b$ and

$$y \ge b - 1 + \frac{\ln(R/(R-1))}{\ln(b/(b-1))}.$$

Let

$$\gamma := \frac{b}{b-1}, \qquad m := \min\{y - b + 1, b\}, \qquad p_{\text{cheap}} := \min\{G(m), 1\}.$$

Then $p_{\text{cheap}} = 1$. Indeed, if $m = y - b + 1 < b$, then the above assumption implies

$$m \ge \frac{\ln(R/(R-1))}{\ln \gamma},$$

hence

$$\gamma^m \ge \frac{R}{R-1}, \qquad G(m) = (R-1)(\gamma^m - 1) \ge 1.$$

If instead $m = b$, then $G(m) = G(b) \ge 1$ follows from Step 1, namely from the assumption that the feasible set is nonempty.

Therefore $p_{\text{cheap}} = 1$.

Since $p_{\text{cheap}} = 1$, the defining condition becomes

$$p^\star = \min\Big\{p \in [0, 1] : \Phi_y(p) \ge \max\{\Delta_y, \Phi_y(1)\}\Big\}.$$

Because $F_1(x) = \min\{G(x), 1\}$ is feasible by Step 1 together with the fact that $G(b) \geq 1$, we have

$$\Phi_y(1) \geq \Delta_y,$$

and hence

$$p^\star = \min\Big\{p \in [0,1] : \Phi_y(p) \geq \Phi_y(1)\Big\}.$$

Moreover, $\Phi_y$ is strictly increasing on $[0,1]$: on each interval $[G(k-1), G(k)]$ with $1 \leq k \leq b$,

$$\Phi_y(p) = \sum_{x=1}^{k-1} G(x) + (y - k + 1)p,$$

whose slope is $y - k + 1 \geq y - b + 1 > 0$. Therefore the only $p \in [0,1]$ satisfying $\Phi_y(p) \geq \Phi_y(1)$ is $p = 1$. Hence $p^\star = 1$. Consequently, the optimal CDF reduces to

$$F^\star(x) = \min\{G(x), 1\} = \min\left\{(R-1)\left(\left(\frac{b}{b-1}\right)^x - 1\right), 1\right\}, \qquad x \geq 0.$$

This is consistent with the optimal CDF in Theorem 5.3.

## D. Algorithm Design Procedure for Water-Filling

This section provides additional intuition and a derivation-oriented view of Algorithm 2 (Water-Filling for Randomized Robust Ski Rental). The key message is that the algorithm is a discrete analogue of the standard *water-filling / thresholding* principle for linear programs: for a candidate water level $h$, we allocate probability mass only on indices with $g(t) \leq h$, and we allocate it *as much as possible* while maintaining $R$-robustness. The minimum feasible $h$ yields a decent solution.

### D.1. Step 1: Computing the cost function $g$

Fix a predicted distribution $p$ over $D \in \mathbb{N}$ and a buy cost $b \in \mathbb{N}$. For the (deterministic) threshold rule that buys on day $t \in \mathbb{N}$, its expected cost under $p$ is

$$g(t) = \sum_{d < t} p(d)\, d \;+\; \sum_{d \geq t} p(d)\, (b + t - 1). \tag{23}$$

We will use the structure of $g(\cdot)$ to decide where probability mass should be placed in the randomized policy.

Let the (finite) support of $p$ be $\{d_1 < d_2 < \cdots < d_n\}$ with probabilities $p(d_i) = p_i$. For any integer $t$ satisfying $d_k < t \leq d_{k+1}$ (with the convention $d_0 := 0$), the event $\{D < t\}$ equals $\{d_1, \ldots, d_k\}$ and $\{D \geq t\}$ equals $\{d_{k+1}, \ldots, d_n\}$. Substituting into (23) yields

$$\begin{aligned}
g(t) &= \sum_{i=1}^{k} p_i d_i \;+\; \Big(\sum_{j=k+1}^{n} p_j\Big)(b + t - 1) \\
&= a_k\, t \;+\; b_k, \qquad \text{for } d_k < t \leq d_{k+1},
\end{aligned} \tag{24}$$

where

$$a_k := \sum_{j=k+1}^{n} p_j = \Pr[D > d_k], \qquad b_k := \sum_{i=1}^{k} p_i d_i + (b-1)a_k. \tag{25}$$

Hence $g(\cdot)$ is (discrete) piecewise linear with slopes $a_k \in [0,1]$ satisfying $a_0 \geq a_1 \geq \cdots \geq a_n = 0$. In particular, $g(t) = t + (b-1)$ for $1 \leq t \leq d_1$, and $g(t) \equiv \mathbb{E}[D]$ for $t > d_n$.

### D.2. Step 2: Why a water level $h$ and why binary search?

We seek a randomized stopping distribution $f$ that minimizes the linear objective

$$\min_f \sum_{t \geq 1} g(t)\, f(t) \tag{26}$$

subject to the $R$-robustness constraints in Lemma 5.1. This is a linear program in the variables $\{f(t)\}_{t \geq 1}$. A standard way to understand optimal solutions of such LPs is through a *threshold/water-level* principle: for an optimal solution there exists a scalar $h$ such that (informally) we only place probability mass on indices with $g(t)$ at or below $h$, and all indices strictly below $h$ are "filled as much as possible" until a constraint becomes tight.

Algorithm 2 implements this idea explicitly: given a candidate level $h$, it constructs the *maximal* feasible allocation of probability mass on the region $\{t : g(t) \leq h\}$ (maximal in the sense that any additional mass within $\{g \leq h\}$ would violate robustness). If this maximal allocation still does not reach total mass 1, then no feasible solution can achieve objective value $\leq h$, so $h$ is infeasible. Conversely, if we can reach total mass 1 while keeping all mass on $\{g \leq h\}$, then $h$ is feasible. Therefore the optimal objective value equals the minimum feasible $h$, which can be found by binary search.

Concretely, on a segment where $g(t) = a_k t + b_k$ and $a_k > 0$, the set of integer days with $g(t) \leq h$ is an interval. The last feasible day inside $(d_k, d_{k+1}]$ is

$$e_k = \min\left\{ \left\lfloor \frac{h - b_k}{a_k} \right\rfloor, d_{k+1} \right\}. \tag{27}$$

Thus, for a fixed $h$, the "active region" $\{t : g(t) \leq h\}$ is a union of intervals in $t$, and we can fill those intervals from left to right.

### D.3. Step 3: Why the geometric update on $[1, b-1]$?

The robustness constraints for $x \leq b - 1$ in Lemma 5.1 are

$$\mu(x) + (b - x)F(x) \leq (R - 1)x, \qquad 1 \leq x \leq b - 1, \tag{28}$$

where

$$F(x) = \sum_{t=1}^{x} f(t), \qquad \mu(x) = \sum_{t=1}^{x} (t - 1)f(t).$$

A guiding principle in the water-filling construction is: *on indices where we actively place mass (i.e., $f(x + 1) > 0$), the constraint* (28) *should be tight.* If (28) had slack at some $x$ while we still place mass at a later day, then we could shift a small amount of probability mass from $x + 1$ to $x$, decrease the objective (since $g$ is increasing on active ranges), and maintain feasibility—contradicting maximality/optimality. This is the same "tightness on the support" phenomenon used in Theorem 5.2.

Assuming tightness at $x$ and $x + 1$,

$$\mu(x) + (b - x)F(x) = (R - 1)x, \qquad \mu(x + 1) + (b - x - 1)F(x + 1) = (R - 1)(x + 1),$$

and using the discrete identities

$$F(x + 1) = F(x) + f(x + 1), \qquad \mu(x + 1) = \mu(x) + xf(x + 1),$$

we obtain the recurrence

$$(b - 1)\big(F(x + 1) - F(x)\big) - F(x) = R - 1, \qquad (x < b \text{ on the tight region}). \tag{29}$$

Equivalently,

$$F(x + 1) = F(x) + \frac{F(x) + R - 1}{b - 1}, \tag{30}$$

which is exactly the geometric-growth update used in Algorithm 2. Thus, the geometric evolution is not an ad-hoc choice: it is the unique way to keep (28) tight while adding mass as aggressively as possible.

### D.4. Step 4: Why do we add an atom across gaps?

Because $g(\cdot)$ is determined by the predicted distribution $p$, the active set $\{t : g(t) \leq h\}$ may skip an interval of days. When there is a gap $(e, d)$ with no active days (so we set $f(t) = 0$ for $e < t < d$), we have $F(t) \equiv F(e)$ and $\mu(t) \equiv \mu(e)$ throughout the gap. If the next active point is $d$ (typically a support point of $p$ where the slope of $g$ changes), we can only increase $F$ by placing an *atom* at $d$.

The atom size is determined again by enforcing the tightness of the robustness constraint. Let $m$ be the probability mass placed at day $d$, so

$$F(d) = F(e) + m, \qquad \mu(d) = \mu(e) + (d-1)m.$$

Imposing tightness at $x = d$ in (28) gives

$$\mu(e) + (d-1)m + (b-d)(F(e) + m) = (R-1)d,$$

and using tightness at $x = e$ (i.e., $\mu(e) + (b-e)F(e) = (R-1)e$) yields

$$m = \frac{(R-1+F(e))(d-e)}{b-1}. \tag{31}$$

This is precisely the gap mass update in Algorithm 2. Intuitively, when the cost level $h$ forces us to skip some days, the only way to keep accumulating probability mass while respecting robustness is to jump via atoms at the next admissible indices, and the jump size is uniquely pinned down by tightness.

### D.5. Step 5: Why a separate tail feasibility test for $x \geq b$?

Lemma 5.1 shows that for $x \geq b$ the robustness requirement is fully captured by the single tail moment constraint

$$\mu(\infty) = \sum_{t \geq 1}(t-1)f(t) \leq (R-1)b. \tag{32}$$

After the water-filling pass constructs $F$ and $\mu$ on $[1, b-1]$ for a candidate $h$, we are left with a remaining probability mass

$$m_{\text{tail}} := 1 - F(b-1)$$

and a remaining moment budget

$$B_{\text{tail}} := (R-1)b - \mu(b-1).$$

Any completion of $f$ on $t \geq b$ must satisfy

$$\sum_{t>b} f(t) = m_{\text{tail}}, \qquad \sum_{t>b}(t-1)f(t) \leq B_{\text{tail}}, \qquad f(t) \geq 0. \tag{33}$$

Given $h$, we additionally require that all mass is placed on indices with $g(t) \leq h$. Therefore, $h$ is feasible if and only if there exists at least one tail day $t \geq b$ with $g(t) \leq h$ that is compatible with the moment budget (33). In particular, if we place all remaining mass at a single day $t$ (which is optimal for feasibility because it minimizes the required moment among choices with fixed support), the moment constraint becomes

$$m_{\text{tail}}(t-1) \leq B_{\text{tail}} \quad \Longleftrightarrow \quad t \leq 1 + \frac{B_{\text{tail}}}{m_{\text{tail}}}.$$

Thus, a simple sufficient-and-necessary feasibility check is:

$$\exists\, t > b \text{ with } g(t) \leq h \text{ and } t \leq 1 + \frac{(R-1)b - \mu(b-1)}{1 - F(b-1)}. \tag{34}$$

This is the tail test used in Algorithm 2 (up to the equivalent indexing convention for $(t-1)$ in $\mu$). When (34) holds, we can complete the distribution by assigning the remaining mass to an admissible tail point within the budget; otherwise, no allocation restricted to $\{g \leq h\}$ can satisfy robustness, and we must increase $h$.

### D.6. Summary of the design

Putting the pieces together, Algorithm 2 follows a consistent logic:

1. Compute the piecewise linear structure of $g(\cdot)$ (Step D.1).

2. For a candidate water level $h$, fill mass on $\{g \leq h\}$ as much as possible on $[1, b-1]$ by keeping robustness constraints tight, which forces the geometric update (30) and the atom rule (31) (Steps D.3–D.4).

3. Check whether the remaining mass can be placed in the tail without violating the moment budget (32) while staying in $\{g \leq h\}$ (Step D.5).

4. Binary search on $h$ to find the minimum feasible level, which yields an optimal robust policy minimizing $\sum_t g(t) f(t)$ under the predicted distribution.

This provides a principled rationale for each component of the algorithm: the water level $h$ represents the optimal objective value, tightness enforces maximal allocation under robustness, and the tail constraint reduces to a one-dimensional moment feasibility check.

## D.7. Implementation-Level Description of Water-Filling Algorithm

---

**Algorithm 3** Water-Filling for Randomized Robust Ski Rental

---

**Require:** Prediction $p$ supported on $\{d_1, \ldots, d_n\}$, buy cost $b$, robustness $R$, tolerance $\epsilon$
**Ensure:** PMF $f$ (and CDF $F$)

1: Compute piecewise linear cost function $g(t) = a_k t + b_k$ for segments $(d_k, d_{k+1}]$.
2: Initialize bounds $h_{\min} \leftarrow 0$, $h_{\max} \leftarrow \max(g(t))$, and optimal level $h^* \leftarrow h_{\max}$.
3: **while** $h_{\max} - h_{\min} > \epsilon$ **do**
4:     Set candidate water level $h \leftarrow (h_{\min} + h_{\max})/2$.
5:     Initialize $F \leftarrow 0$, $\mu \leftarrow 0$, feasible $\leftarrow$ FALSE.
6:     **for** each segment $k$ where interval starts before $b$ **do**
7:         Find active range $[s, e]$ in $(d_k, d_{k+1}]$ where $g(t) \leq h$:

$$e = \min \left\{ \left\lfloor \frac{h - b_k}{a_k} \right\rfloor, \; d_{k+1}, \; b \right\}.$$

8:         **if** $s \leq e$ **then**
9:             **1. Atom Update (Gap Filling):** Check slack at start $s$.
10:           **if** $(R-1)s > \mu + (b-s)F$ **then**
11:             Add mass $m = \frac{(R-1)s - (\mu + (b-s)F)}{b-1}$ to $F$.
12:             Update $\mu \leftarrow \mu + m \cdot s$.
13:           **end if**
14:             **2. Geometric Fill:** Saturate constraints on $(s, e]$.
15:             Update $F$ using closed-form geometric series:

$$F \leftarrow (F + R - 1) \left( 1 + \frac{1}{b-1} \right)^{e-s} - (R-1).$$

16:             Update $\mu \leftarrow (R-1)e - (b-e)F$.
17:         **end if**
18:         **if** $F \geq 1$ **then**
19:             feasible $\leftarrow$ TRUE.
20:             **break** loop.
21:         **end if**
22:     **end for**
23:     **if** feasible is FALSE **then**
24:         **3. Tail Feasibility** ($x \geq b$)**:**
25:         Calculate remaining budget $B_{\text{tail}} = (R-1)b - \mu$.
26:         **if** $\exists d_j \geq b$ such that $g(d_j) \leq h$ and $d_j \leq \frac{B_{\text{tail}}}{1-F}$ **then**
27:             feasible $\leftarrow$ TRUE.
28:         **end if**
29:     **end if**
30:     **4. Binary Search Update:**
31:     **if** feasible is TRUE **then**
32:         $h^* \leftarrow h$, $h_{\max} \leftarrow h$ {Feasible, try lower cost}
33:     **else**
34:         $h_{\min} \leftarrow h$ {Infeasible, need higher budget}
35:     **end if**
36: **end while**
37: **return** Distribution constructed with $h^*$.

---

**D.8. Time Complexity Analysis**

Let $n := |\text{supp}(p)|$ where $\text{supp}(p) = \{d_1, \cdots, d_n\}$ with $d_1 < \cdots < d_n$, and let $K$ denote the number of piecewise-linear segments of $g(t)$ (here $K = n - 1$ for intervals $(d_k, d_{k+1}]$).

**Precomputation.**    Constructing the segment coefficients $\{(a_k, b_k)\}_{k=1}^K$ and computing $h_{\max} = \max_t g(t)$ over relevant endpoints takes $O(K) = O(n)$ time and $O(K) = O(n)$ space.

**Cost of a feasibility check (inner loop).**    For a fixed candidate level $h$, the algorithm scans segments whose intervals start before the buy cost $b$. Each visited segment performs only $O(1)$ arithmetic operations: (i) computing the active endpoint $e = \min\{\lfloor (h - b_k)/a_k \rfloor, d_{k+1}, b\}$, (ii) a constant-time slack/atom update, and (iii) a constant-time *closed-form* geometric update for $F$ (hence no dependence on the interval length $e - s$ beyond exponentiation). Thus, the segment scan costs $O(K) = O(n)$ in the worst case, with early stopping when $F \geq 1$.

The tail feasibility check ($x \geq b$) can be implemented in $O(n)$ time by scanning the remaining support points $\{d_j \geq b\}$ and testing the two conditions $g(d_j) \leq h$ and $d_j \leq B_{\text{tail}}/(1 - F)$. (With additional preprocessing (e.g., sorting tail points by $g(d_j)$ and maintaining a pointer), this step can be reduced to $O(\log n)$ per check, but the overall worst-case bound below remains dominated by the segment scan.)

Therefore, each feasibility test runs in

$$T_{\text{check}}(n) = O(n) \quad \text{time}, \qquad S_{\text{check}}(n) = O(1) \text{ extra space (beyond stored segments/support)}.$$

**Binary search iterations.**    The outer loop is a standard binary search over $h \in [h_{\min}, h_{\max}]$ with tolerance $\epsilon$, requiring

$$I = \left\lceil \log_2 \left( \frac{h_{\max} - h_{\min}}{\epsilon} \right) \right\rceil$$

iterations.

**Total complexity.**    Combining the above, the overall running time is

$$T_{\text{total}}(n, \epsilon) = O(n) + I \cdot O(n) = O\left( n \log \frac{h_{\max} - h_{\min}}{\epsilon} \right) = O\left( n \log \frac{\max(g(t))}{\epsilon} \right).$$

and the memory usage is $O(n)$ to store the support and segment coefficients (plus $O(1)$ working memory during the checks).

