# OpenReview forum: "Robust and Consistent Ski Rental with Distributional Advice"
_ICML.cc/2026/Conference — ICML 2026 regular_

### Official Review · Reviewer_u52n · 2026-03-04

**Soundness:** 3
**Presentation:** 3
**Significance:** 3
**Originality:** 3
**Overall Recommendation:** 5
**Confidence:** 3

**Summary:**

The authors study the learning augmented ski rental problem. The novel aspect in this work is that they consider a prediction type that is a distribution over days rather than a single day.

The first contribution of the paper is an algorithm that determines the optimal buying threshold for the case when the predictions are perfect and give the true distribution of skiing days. They also provide bounds on the expected cost in this case.

Then they design a "clamp" policy that guards against bad predictions. This clamp policy achieves the usually desired robustness and consistency guarantees of learning augmented algorithms.

A third main contribution is on the domain of randomized algorithms. The algorithm guesses (using binary search) a level h and assigns non-zero probability to days that have a consistency cost at most h. The probability is assigned in a way that ensures the robustness guarantee. If it is not possible to satisfy the constraints, the level h is revised.

Finally the authors conduct an experimental evaluation that shows improvements over the solutions that utilize point predictions.

**Compliance With Llm Reviewing Policy:**

Affirmed.

**Final Justification:**

I maintain my recommendation to accept the paper. The authors study an interesting extension to a fundamental problem and provide a rigorous theoretical analysis.

During the rebuttal, the authors clarified how the water-filling algorithm works to my satisfaction. I also considered the discussion with Reviewer Czvm. The authors agreed to narrow their novelty claims to focus specifically on the ski rental problem rather than all distributional predictions. This limits the originality of the paper, but the specific technical contributions are still non-trivial.

The authors also corrected the technical bugs found in the appendices by Reviewer ns15. Finally, while the empirical validation is purely synthetic, the paper successfully achieves the desired robustness and consistency guarantees and provides a solid contribution to learning augmented algorithms.

**Key Questions For Authors:**

-

**Limitations:**

-

**Strengths And Weaknesses:**

Strengths:
- The theoretical analysis is rigorous and seems correct.
- The paper is mostly well written.
- Studies an interesting extension to a fundamental problem

Weaknesses:
- The empirical validation is purely synthetic.
- I found the description of the water-filling algorithm was somewhat hard to understand.

---

> ### Author Rebuttal · Authors · 2026-03-28
>
> **Response to Reviewer u52n**
>
> We thank the reviewer for the encouraging comments. We are glad that the reviewer found the problem important and the theoretical analysis rigorous. Below we respond to the two concerns.
>
> 1. Empirical validation on synthetic data only.
> We agree that real-world validation would strengthen the practical claims of the paper. Our focus on synthetic distributions (uniform, discretized Gaussian, geometric, bimodal) allowed us to systematically isolate how specific distributional shapes and Wasserstein perturbations affect the trade-off between consistency and robustness. In real-world datasets, these effects are often mixed with other confounding factors and therefore harder to interpret cleanly. In the revision, we will discuss this limitation more explicitly and, if space permits, include additional real-world validation.
>
> 2. Clarity and necessity of the Water-Filling algorithm.
> Thank you for this feedback. We agree that the role of the Water-Filling algorithm should be explained more clearly, and we will revise Section 5.3 accordingly.
>
> - Why it is needed: Unlike point-prediction models, distributional advice gives a full cost landscape rather than a single suggested stopping day. In the randomized setting, we therefore need to optimize over a stopping distribution $f(t)$ that places more mass on favorable buying days while still satisfying the robustness constraints. This is exactly the constrained optimization problem that the Water-Filling algorithm is designed to solve.
>
> - Why it is useful: It reduces a complex high-dimensional search over all possible stopping distributions into a simple, efficient 1D binary search over a single target consistency level $h$. This makes the method both computationally manageable and adaptable to multi-modal or geometric predictions.
>
> - High-level idea:
>   1. Define the Terrain. We first compute the consistency cost function $g(t)$ induced by the predicted distribution. This gives a piecewise-linear “terrain” over possible buying days, where lower values of $g(t)$ correspond to more favorable stopping choices.
>
>   2. Set the Water Level. We then choose a candidate level $h$ and identify the active region consisting of all days with $g(t)\le h$. Intuitively, these are the days whose consistency cost is low enough to be considered at the current level.
>
>   3. Fill the Mass. We place probability mass only on this active region, assigning as much mass as possible while respecting the robustness constraints. On prefixes where these constraints are tight, the resulting CDF follows the geometric update rule.
>
>   4. Check Feasibility and Search for the Best Level. If the total mass can reach $1$ while satisfying all robustness constraints, then the current level $h$ is feasible; otherwise it is infeasible. We then use binary search to find the minimum feasible level $h$, which yields the most consistency-oriented stopping distribution subject to robustness.
>
>   5. Geometric Update. On days where the robustness constraint is tight, the CDF follows the geometric update rule derived in our analysis, capturing the trade-off between consistency and robustness.
>
>
> We will revise Section 5.3 to make this intuition more explicit, and we expect this will make the algorithm slightely easier to follow.

---

> > ### Author Rebuttal · Reviewer_u52n · 2026-04-02
> >
> > Thank you, I keep the original positive score after the clarifications.

---

> > > ### Author Response · Authors · 2026-04-06
> > >
> > > Thank you very much for your constructive engagement throughout the review process. As discussed, we will ensure that the revised version of the paper includes an expanded intuitive explanation of the Water-Filling algorithm to improve clarity, along with a dedicated discussion of the limitations of our synthetic empirical validation. Thank you again for your time and for helping us improve the presentation of our work.

---

### Official Review · Reviewer_Czvm · 2026-03-12

**Soundness:** 3
**Presentation:** 3
**Significance:** 2
**Originality:** 2
**Overall Recommendation:** 3
**Confidence:** 3

**Summary:**

The paper studies the ski rental problem, a well-studied and canonical problem in decision-making under uncertainty, when distributional predictions regarding the usage duration are present. Specifically, the paper aims to design policies that leverage the full distribution of usage duration to minimize expected cost, while providing theoretical guarantees on performance even when predictions are imperfect. This is in contrast to recent work in the space of learning-augmented algorithms, which integrate only point predictions in such decision-making problems under uncertainty.

The results of the paper are the following:

First, for deterministic algorithms, they formalize the distributional ski rental setting with perfect prediction. For threshold-based policies (i.e., when to buy), they propose an efficient $O(n)$ (minor note: $n$ I assume is an upper bound on the size of the support of the distribution. It is not mentioned here where the notation and the result is introduced, so it’s at first a bit confusing) algorithm to compute the optimal threshold. The objective to evaluate performance is a standard (expected) competitive ratio in this case. They then address the standard scenario in learning-augmented algorithms in which the prediction can be imperfect and an algorithm should navigate the robustness-consistency tradeoff. They propose a clamp policy for the optimal threshold, and show that it has these two desirable properties: (i) it doesn’t degrade arbitrarily with prediction quality and (ii) it approaches the optimal value as the error vanishes.

Then, they provide results for randomized algorithms as well. First, they formulate necessary robustness constraints for randomized policies, deriving conditions on the CDF of the duration that ensure a bounded competitive ratio. They also characterize the stopping distribution via a Water-Filling Algorithm, which optimizes expected cost while strictly satisfying robustness constraints.

Finally, they provide some experiments (with synthetic data) against two baselines, where they showcase that consistency is better than the baselines while achieving the same robustness.

**Compliance With Llm Reviewing Policy:**

Affirmed.

**Final Justification:**

I appreciate the authors' willingness to present the novelty claim more precisely and their efforts to correct the technical mistakes identified by reviewer ns15. The paper is well-written, and the results are fairly complete. Thus, I have increased my score. At the same time, I'm still hesitant due to the framing of its contributions. Since it's now established that there was prior recent work studying distributional predictions for learning-augmented algorithms, making the initial framing slightly misleading, it's also not the first paper to study distributional predictions in the context of the ski-rental problem. As noted, Diakonikolas et al. (2021) study a different model with distributional predictions; also, Besbes et al. [1] study a similar setting (as I understand, also with a different approach). In my opinion, some further discussion would be needed here to better justify the merits of this submission. However, since the other reviewers are more positive than me, I won't push against acceptance during the discussion phase.


[1] O. Besbes, W. Ma, and O. Mouchtaki, Beyond IID: data-driven decision making in heterogeneous environments. *Management Science* (2025)

**Key Questions For Authors:**

My main question relates to what I stated as the main weakness, namely, why those papers using distributional information for learning-augmented algorithms are not cited, and what the connections of the current paper are with them.

In 5.1, what do these conditions for robustness tell us intuitively? In general, section 5 has all these statements without any further explanations, making it hard to appreciate the contribution. I understand the space limitations, that’s why I’d ask the authors here to explain a bit more the steps and intuition behind the results in sections 5.1 and 5.2.

**Limitations:**

yes

**Strengths And Weaknesses:**

**Strengths**

It is a well-written paper, where the results are stated clearly and the overall presentation is coherent. It also does a good job in terms of clarity of notation and proofs. Motivation is indeed good, as having some (imperfect) distributional advice at hand for parameters of the problem seems more realistic than point predictions. The paper provides a satisfactory answer both for deterministic and for randomized algorithms.

The synthetic experiments and the results presented follow the standard style of presentation in the literature of learning-augmented algorithms. Nevertheless, I find it a strength of the paper that they put the effort to also compare their algorithms against baselines, which have to be modified to be adjusted to the case of distributional predictions, and allow for a fair comparison.

**Weaknesses**

My main issue is with how the paper frames its contribution wrt being the first to leverage distributional information for the predictions in learning-augmented algorithms. This is how it’s understood in the abstract/intro, and at the end of 1.2 the authors write: “Consequently, our work is the first to simultaneously leverage distributional information while providing a robustness guarantee”. But is this true? What about this ICML paper ​​https://proceedings.mlr.press/v139/diakonikolas21a/diakonikolas21a.pdf which is not cited, what’s the relation to it? Because they also use the same offline benchmark, i.e., an optimal strategy that knows the distribution in advance. They just draw samples from this distribution instead of assuming full knowledge (which is arguably a more realistic assumption) and also study the ski-rental problem. I guess the authors don’t consider that this paper models distributional, but rather point predictions? Maybe there’s something I’m missing or misunderstood here. There are also other papers that consider online algorithms with distributional predictions, such as https://arxiv.org/abs/2505.04949, https://arxiv.org/abs/2510.25582  or, more importantly, https://arxiv.org/abs/2411.16030 (where they actually state in the abstract: We initiate the study of algorithms with distributional predictions, where the prediction itself is a distribution), but none of these papers is cited here.
I’d appreciate it if the authors elaborate on the comparison with the above papers. If it’s not the first paper to introduce this idea of distributional predictions as a middle ground, then it loses a lot of its conceptual contribution to the field.

Not a significant weakness, but the algorithms designed in both settings follow fairly standard techniques and ideas of the field. Restricting the threshold to an interval to simultaneously achieve robustness and consistency is an expected approach to balance the two. Also, water-filling-style algorithms for randomized cases have appeared before. Given that the rest of the techniques and results are specific to the ski rental problem, I’m not sure if and how the technical contributions of this paper can be used and extended to other problems of decision-making under uncertainty.

Minor typo: You say you use $P$ to denote probability in 2.5. And right afterwards, you use $\Pr$.

---

> ### Author Rebuttal · Authors · 2026-03-27
>
> ### Response to Reviewer Czvm
>
> We sincerely thank the reviewer for the thoughtful critique and for pointing us to these highly relevant papers. We intended the claim of being the first only in the context of the  ''ski rental problem''. While our original draft already cited prior work on distributional predictions for binary search [arXiv:2411.16030], we did not make this limitation sufficiently explicit, and we apologize for the resulting confusion. In the revision, we will narrow this statement and instead say that our work provides a systematic treatments of ski rental with full distributional advice under an untrusted-advisor framework.
>
> Below, we clarify the distinctions between our work and the literature highlighted by the reviewer.
>
> 1. Distinction from Diakonikolas et al. (2021)
>
> While both papers involve ski rental and distributions, they study fundamentally different models and objectives.
>
> Diakonikolas et al. (Honest Oracle, Stochastic Reality): This work adopts a PAC-learning style framework in which the future is drawn from an unknown distribution and the algorithm receives i.i.d. samples from that underlying distribution. The goal is to learn a near-optimal policy in a stochastic environment.
>
> Our Work (Untrusted Oracle, Potentially Adversarial Reality):  We follow the learning-augmented paradigm where the future outcome may be adversarial, and the algorithm is given a predictive distribution $\hat{p}$ from an untrusted source. Our central challenge is achieving $R$-robustness even when $\hat{p}$ is inaccurate or maliciously misleading. This robustness-against-untrusted-advice aspect is not addressed in a sampling-based stochastic framework.
>
> 2. Relation to recent work on distributional predictions
>
> We appreciate the reviewer’s pointers to the growing literature on distributional advice, including recent work such as [arXiv:2505.04949], [arXiv:2510.25582].
> These works share our motivation for leveraging richer predictive signals in matching and bidding.
>
> However, the technical difficulty in ski rental lies in the unique interaction between a rent-or-buy stopping decision, hard robustness constraints, and randomized policies. Our analysis relies on structural properties specific to this cost-balance, rather than directly following techniques from matching or bidding domains. We will also cite those works mentioned by the reviewer in revision.
>
>
> 3. Intuition for Section 5 (Robustness Conditions).
>
> The reviewer requested intuition behind the conditions in 5.1 and 5.2. Intuitively, these conditions dictate how the probability mass of buying must be distributed over time to ensure that the expected competitive ratio never exceeds $R$, regardless of the true adversarial stopping time.
>
> Section 5.1 (Early Stopping): The conditions force the algorithm to maintain a tight geometric growth in the cumulative distribution function (CDF) to actively "hedge" against early terminations.
>
> Section 5.2 (Tail Behavior): The condition relaxes into a moment bounding constraint, effectively limiting the maximum expected buying day if the request continues indefinitely.
> This step-by-step bounding of the expectation is what ultimately allows our water-filling algorithm to safely maximize consistency with the prediction without violating worst-case guarantees. We will add this explanation to the paper to make the results in Section 5 much more accessible.
>
>
>
>  4. On novelty and technical contribution
>
> We respectfully submit that the results go beyond a straightforward application of existing ideas. While principles such as thresholding and water-filling are classical, applying them to an untrusted distribution uncovers a nontrivial and, to our knowledge, previously uncharacterized phase transition in the optimal randomized policy. Depending on the available robustness slack, the optimizer either places mass on an active geometric prefix to hedge against early risk, or skips that prefix entirely and places a delayed atom in the tail. Identifying this “prefix-skipping” behavior requires a careful analysis of the robustness-consistency tradeoff that goes beyond existing point-prediction methods. This phenomenon is a direct result of the interaction between robustness and consistency in the ski-rental cost structure and does not follow as a routine corollary of existing distributional-advice results. We believe these insights offer a useful structural blueprint for extending distributional advice to a broader class of online covering problems.
>
>
>  We will revise our paper to present our novelty claim in this more precise and appropriately scoped way.

---

> > ### Author Rebuttal · Reviewer_Czvm · 2026-04-04
> >
> > Thank you for the detailed response. I appreciate the comparison provided and the willingness to reframe the contribution of the paper. I am willing to update my score to weak reject, because I still think that this narrower contribution of the paper to its subfield (i.e., learning-augmented algorithms with distributional predictions) weakens the novelty it initially claimed. Now, it might take a bit more effort to justify that we actually need to understand this new model very well in the context of the ski-rental problem, with the potentially adversarial reality (the untrusted oracle is probably easier to justify). This, together with the bugs in the proofs found by ns15, makes it difficult for me to update my score in the accept territory for the initial submission.

---

> > > ### Author Response · Authors · 2026-04-06
> > >
> > > Thank you for your continued engagement and for carefully considering our rebuttal. We appreciate your clarification and your  willingness to revisit your score. We understand your perspective regarding the scope of our novelty, and we will make sure the revised version more carefully scopes the contribution.
> > >
> > > Furthermore, regarding the mathematical errors identified by Reviewer ns15, we would like to clarify that these parts were not only corrected, but substantially reworked through a more rigorous proof architecture, which Reviewer ns15 has reviewed and considered fully resolved.
> > >
> > > Thank you again for your thoughtful feedback, which has helped us frame the paper more precisely.

---

### Official Review · Reviewer_pW2v · 2026-03-13

**Soundness:** 3
**Presentation:** 3
**Significance:** 2
**Originality:** 2
**Overall Recommendation:** 4
**Confidence:** 3

**Summary:**

The paper studies learning-augmented ski rental with distributional predictions rather than point predictions. It provides a framework for designing deterministic and randomized algorithms that use the full predicted distribution while preserving worst-case robustness. For deterministic algorithms, it gives an optimal threshold rule under perfect prediction and a policy with robustness and consistency guarantees under prediction error. For randomized algorithms, it derives an algorithm that optimizes the expected cost while strictly satisfying robustness constraints. The experimental results suggest that these distribution-aware approaches yield better consistency than adapted point-prediction baselines across several representative distributions.

**Compliance With Llm Reviewing Policy:**

Affirmed.

**Key Questions For Authors:**

N/A

**Strengths And Weaknesses:**

**Strengths:**
1. The ski rental problem is a fundamental challenge in decision-making under uncertainty, with many important applications.

2. This paper is the first to formalize the problem in the distributional-advice setting. I believe this is a step in the right direction, since many machine learning models naturally output a distribution rather than a single point prediction. As the authors demonstrate, leveraging the full distribution can lead to strictly stronger results than those obtainable in the point-prediction setting.

3. By formalizing the problem under perfect predictions, deriving sufficient conditions for competitiveness and robustness, and studying both deterministic and randomized settings, the paper provides a comprehensive treatment of the problem and sets the foundation for future work.

**Weaknesses:**
1. While the paper definitely adds to the literature and takes a meaningful step toward understanding distributional advice, I did not find the results or techniques particularly exciting. The paper is careful and comprehensive, but the overall contribution feels somewhat incremental rather than especially novel or technically deep.

---

> ### Author Rebuttal · Authors · 2026-03-28
>
> **Response to Reviewer pW2v**
>
> We thank the reviewer for the thoughtful feedback. We are glad that the reviewer found the problem fundamental and viewed the paper as a meaningful first step toward understanding distributional advice in ski rental.
>
> We understand the reviewer’s impression that, at the level of problem formulation, moving from point predictions to distributional predictions may feel natural rather than especially surprising. At the same time, we would like to clarify that, on the solution side, this shift leads to a substantially different technical problem and is not just a routine extension of the point-prediction setting.
> In the point-prediction setting, the advice effectively specifies a single target day, so the main question is how to hedge around that point while preserving robustness. In contrast, full distributional advice induces an entire cost landscape over stopping times. This changes the structure of the problem in both the deterministic and randomized settings.
>
> Wasserstein Stability (Deterministic Setting): Bounding the performance of the Clamp Policy under arbitrary distributional shifts (Theorem 4.4) requires mapping functional distances to competitive ratios. To achieve this, we had to establish cost and optimum stability lemmas using Kantorovich-Rubinstein duality (Appendix B.1). This provides a new, rigorous analytic framework for handling continuous prediction errors rather than simple binary "right/wrong" point prediction models..
>
> Complex Constrained Optimization (Randomized Setting): Designing the randomized policy requires optimizing the stopping distribution $f(t)$ to minimize the expected cost against a piecewise-linear consistency cost function, subject to an infinite set of pointwise and tail robustness constraints. As detailed in Section 5 and Appendices C and D, our Water-Filling algorithm does not simply adapt existing point-prediction techniques. Instead, it actively constructs the optimal cumulative distribution by enforcing strict tightness on the support constraints, which dictates the exact geometric update rule and specific atomic gap-fills we derived.
>
> We agree that the paper is intended more as a first systematic treatment of the distributional-advice setting than as a single dramatic theorem. In the revision, we will revise the introduction and the transition into Section 5 to better highlight these new structural and analytical aspects, so that the contribution is positioned more clearly.
>
> We thank the reviewer again for the careful reading.

---

> > ### Author Rebuttal · Reviewer_pW2v · 2026-04-03
> >
> > Thank you for your thorough response. I will maintain my positive score.

---

> > > ### Author Response · Authors · 2026-04-06
> > >
> > > Thank you for your constructive engagement throughout the review process.. We appreciate your recognition of this paper as a meaningful first step toward distributional advice in ski rental. As promised, we will revise the introduction and Section 5 in the final version to more clearly highlight the new structural and analytical ideas needed when moving from point predictions to full distributions.

---

### Official Review · Reviewer_ns15 · 2026-03-13

**Soundness:** 2
**Presentation:** 3
**Significance:** 3
**Originality:** 3
**Overall Recommendation:** 4
**Confidence:** 2

**Summary:**

The paper studies the  problem of ski rental, which has been extensively studied in the field of online algorithms. In particular, we know that the best deterministic algorithm has a competitive ratio of 2 and the best randomized algorithm achieves a competitive ratio of e/(e-1). Due to its simplicity, the problem has been used as a starting point for many new conceptual models in online algorithms, like incorporating predictions or bounding the tail risk of the competitive ratio.

The authors take a nice step and study the problem under distributional assumptions, departing from the, recently, standard approach of assuming access to black-box predictors. In my opinion, this is a very natural and interesting model to study.

They first study the case of threshold algorithms given access to the true distribution of skiing days and then design algorithms that use a predicted distribution of skiing days with the goal of being robust & consistent.

**Compliance With Llm Reviewing Policy:**

Affirmed.

**Final Justification:**

The technical mistakes I spotted in the write-up were correct but they were revised by the authors.

**Key Questions For Authors:**

Could the authors please clarify the above points and lets us know if there are indeed mistakes?

**Limitations:**

yes

**Strengths And Weaknesses:**

Strengths:

- The model is quite natural and is a nice extension of the learning-augmented approach to designing online algorithms. It offers a direct way to “use” statistical knowledge that a practitioner might have for the problem at hand.

- The authors showcase that using distributional knowledge is strictly better than using a point-prediction (which essentially can be seen as some function of the true distribution, e.g. mean, median etc).

Weaknesses:

My main objections concern 2 potential bugs I have found in the paper. I apologize in advance if there is an error on my side.

A) Appendix C.4 / Theorem C.1:

In Step 2 of the proof, lines 1035-1044, I don’t see how equation (12) is  derived from the robustness constraints at x and x+1. It appears that you have subtracted the inequalities(?), but of course we cannot subtract inequalities.

B) Appendix C.3 / Theorem 5.3:
The paper claims in Theorem 5.3 that the geometric cumulative distribution function (CDF) from Theorem 5.2 remains optimal under the "one-drop tail-flat" condition, which includes the one-hot prediction case $\hat{p}(y)=1$ for $y\ge b$. However, this claim appears to not hold. I'm attaching a counterexample demonstrating that the geometric CDF is suboptimal in this regime:

Consider the expected cost of the deterministic threshold policy $g(t)$ under the predicted distribution $\hat{p}$. For the one-hot prediction $y\ge b$, the cost function is: $g(t)=b+t-1$ for $t\le y$ and  $g(t)=y$ for $t\ge y+1$ Let us instantiate this with $b=5$, $R=2$, and a one-hot prediction exactly at day 5 ($\hat{p}(5)=1$, so $y=b=5$). This yields: $g(t)=5+t-1$ for $t\le 5$ and $g(t)=5$ for $t\ge 6$.

With $b=5$ and $R=2$, the CDF evaluates to $F^{*}(x)=\min((1.25)^{x}-1,1)$. This generates the following probabilities:
- $F^{*}(1)=0.25\implies f(1)=0.25$

- $F^{*}(2)=0.5625\implies f(2)=0.3125$
- $F^{*}(3)=0.953125\implies f(3)=0.390625$
- $F^{*}(4)=1\implies f(4)=0.046875$

The expected objective value for this distribution is: $5(0.25)+6(0.3125)+7(0.390625)+8(0.046875)=6.234375$

Consider instead the deterministic policy "buy on day 6 with probability 1" ($f(6)=1$). The expected cost is exactly $g(6)=5$. This is strictly lower than the claimed optimal cost of $6.234375$.

Feasibility Check: This policy rigorously satisfies the robustness constraints set out in Lemma 5.1. For $x\in\{1,2,3,4\}$, we have $F(x)=0$ and $\mu(x)=0$ , meaning the pointwise constraints hold trivially. For the tail constraint, $\mu(\infty)=6-1=5$. Since $(R-1)b=(2-1)5=5$, the constraint $\mu(\infty)\le(R-1)b$ is satisfied.

---

> ### Author Rebuttal · Authors · 2026-03-29
>
> Thank you for these careful observations. Both points are well taken. Appendix C.4 in the submitted version contained an invalid derivation, and Theorem 5.3, as stated, requires revision. We have revised both accordingly. Since we cannot share the revised manuscript during the rebuttal period, we summarize the corrected statements and main proof ideas below.
>
> (A) Appendix C.4 / Theorem C.1. You are right that Eq. (12) in the submitted version was not justified: it came from subtracting the robustness inequalities at $x$ and $x+1$, which is not a valid step. We have deleted that argument completely.
>
> More importantly, we replaced the old theorem by a stronger one. The submitted Theorem C.1 only covered the one-hot objective for $1 \le y \le b-1$. The revised Theorem C.1 gives the exact optimizer for the one-hot objective for every $y \ge 1$, assuming only that the feasible set in Lemma 5.1 is nonempty.
>
> The new proof does not compare two inequalities by subtraction. Instead it starts from a single valid reformulation of the pointwise robustness constraint:
> $$
> H(x)=bF(x)-\sum_{u=1}^x F(u)\le (R-1)x,
> $$
> equivalently
> $$
> (b-1)F(x)\le (R-1)x+\sum_{u=1}^{x-1}F(u).
> $$
> From this, an induction yields the universal geometric envelope
> $$
> F(x)\le G(x):=(R-1)(\gamma^x-1), \qquad \gamma=1+\frac{1}{b-1}.
> $$
> For a fixed endpoint level $p$, we then define the capped front-loaded profile
> $$
> F_p(x)=\min(G(x),p),
> $$
> and prove that it is the canonical feasible prefix with endpoint value $p$: among all feasible CDFs with the same endpoint, it maximizes the relevant cumulative prefix sums and therefore minimizes the objective.
>
> The corrected theorem is as follows.
>
> If $1 \le y \le b-1$, define
> $$
> S_y(p)=\sum_{x=1}^y F_p(x),
> $$
> $$
> \Lambda_y(p)=
> \gamma^{\,b-y-1}\frac{(R-1)(y+1)+S_y(p)}{b-1}
> +(R-1)(\gamma^{\,b-y-1}-1),
> $$
> and
> $$
> p^\star=\min(p\in[0,1]:\Lambda_y(p)\ge 1).
> $$
> Then the optimal CDF is
> $$
> F^\star(x)=F_{p^\star}(x) \qquad (0\le x\le y),
> $$
> $$
> F^\star(x)=
> \min\left(
> \gamma^{x-y-1}\frac{(R-1)(y+1)+S_y(p^\star)}{b-1}
> +(R-1)(\gamma^{x-y-1}-1), 1
> \right)
> \qquad (y+1\le x\le b).
> $$
> and $F^\star(x)=1$ for $x\ge b$.
>
> If $y \ge b$, define
> $$
> \Phi_y(p)=(y-b)p+\sum_{x=1}^b F_p(x), \qquad
> \Delta_y=\max(y-(R-1)b,0),
> $$
> $$
> m=\min(y-b+1,b), \qquad p_{\mathrm{cheap}}=\min(G(m),1),
> $$
> and
> $$
> p^\star=\min(p\in[0,1]:\Phi_y(p)\ge \max(\Delta_y,\Phi_y(p_{\mathrm{cheap}}))).
> $$
> Then the optimal CDF is
> $$
> F^\star(x)=F_{p^\star}(x) \qquad (0\le x\le b),
> $$
> $$
> F^\star(x)=p^\star \qquad (b+1\le x\le y),
> $$
> and
> $$
> F^\star(x)=1 \qquad (x\ge y+1).
> $$
>
> Thus the incorrect Step 2 is not merely patched; it is removed and replaced by a different proof architecture. The new argument is based on: (i) the envelope $F\le G$, (ii) the extremality of the front-loaded cap $F_p$, and (iii) an explicit feasibility characterization of tail completion. This gives an exact solution for all $y \ge 1$.
>
> (B) Appendix C.3 / Theorem 5.3. We also agree that the submitted theorem was false as stated. Your one-hot counterexample is valid, and it shows that the geometric CDF from Theorem 5.2 need not remain optimal under the original “one-drop tail-flat” condition. We therefore changed both the theorem statement and its proof.
>
> The submitted theorem assumed: there exists $y \ge b$ such that $g(1)\le \cdots \le g(y)$, $g(y+1)\le g(y)$, and $g(t)=g(y+1)$ for all $t\ge y+1$, and then concluded that the geometric CDF remains optimal. This is the claim that fails.
>
> The revised theorem is different. It now assumes that there exists
> $$
> y\ge b-1+\frac{\ln(R/(R-1))}{\ln(b/(b-1))}
> $$
> such that
> $$
> g(1)\le \cdots \le g(y)
> \qquad \text{and} \qquad
> g(t)\ge g(y+1-b) \quad \text{for all } t\ge y+1.
> $$
> Under this corrected condition, the geometric CDF from Theorem 5.2,
> $$
> F^\ast(x)=\min((R-1)((1+\tfrac{1}{b-1})^x-1), 1),
> $$
> is still optimal.
>
> The intuition is that under the above lower bound on $y$, the geometric CDF has already reached $1$ early enough, so the only remaining requirement is that the far tail is not cheaper than the relevant earlier comparison point $y+1-b$. This is why the revised sufficient condition differs from the original tail-flat statement.
>
> So the revised scope is neither simply weaker nor simply stronger than the original one. In particular, for the one-hot case, geometric optimality is now claimed only when
> $$
> y\ge b-1+\frac{\ln(R/(R-1))}{\ln(b/(b-1))}.
> $$
> For your example $(b,R,y)=(5,2,5)$, this threshold equals
> $$
> 4+\frac{\ln 2}{\ln(5/4)}\approx 7.1,
> $$
> so the example is correctly excluded by the revised theorem. The interval
> $$
> b\le y< b-1+\frac{\ln(R/(R-1))}{\ln(b/(b-1))}
> $$
> is no longer claimed by Theorem 5.3; instead, it is handled exactly by the new all-$y \ge 1$ Theorem C.1 above.
>
> We sincerely thank you for catching both issues. Your comments led us to remove an invalid derivation, correct an incorrect theorem statement, and replace both affected arguments by rigorous revised results whose scopes are now stated precisely.

---

> > ### Author Rebuttal · Reviewer_ns15 · 2026-04-04
> >
> > I thank the authors for their response and I appreciate the effort in correcting their mistakes.
> > I will update my score to weak accept. Please make sure to update the statements and proofs in the final version of the paper.

---

> > > ### Author Response · Authors · 2026-04-06
> > >
> > > Thank you very much for your constructive engagement throughout the review process. We sincerely appreciate your careful reading of the appendices and your recognition of our revisions. We confirm that the new rigorous proof architecture for Theorem C.1, as well as the revised statement and proof of Theorem 5.3 exactly as described in our earlier response, will be fully incorporated into the final version of the paper.

---

### Decision · Program_Chairs · 2026-04-30

**Decision:**

Accept (regular)

**Comment:**

The paper studies the classic ski rental problem in the learning-augmented setting, where the prediction is a distribution instead of a single number.

The reviewers agree that this a natural model to study this problem, and that the paper indeed shows that distributional predictions result in better algorithms than with just one-number predictions.

This is not the first paper to introduce this model, but the first one to study this problem in this model.

There were some bugs in the initially submitted version of the paper, but they seem to be fixed now.

Some of the reviewers consider the paper well written, some others mention that there are parts that are hard to understand.

The reviewers also have mixed opinions regarding the novelty of techniques.